

**Changes in Particulate and Mineral Associated Organic Carbon with**
**Land Use in Contrasting Soils**
**Sabina Yeasmin[1][†][*], Balwant Singh[1], Cliff T Johnston[2], Donald L Sparks[3] and Quan Hua[4]**
[1]Sydney Institute of Agriculture, School of Life and Environmental Sciences, The University of Sydney,
Sydney, NSW 2006, Australia
[2]Crop, Soil and Environmental Sciences, Purdue University, West Lafayette, IN 47907, USA
[3]Department of Plant and Soil Sciences, University of Delaware, Newark, DE 19716, USA
[4]Australian Nuclear Science and Technology Organisation, Locked Bag 2001, Kirrawee DC, NSW
2232, Australia
[†]Current address: Department of Agronomy, Bangladesh Agricultural University, Mymensingh-2202,
Bangladesh.
**\***Corresponding author: sabinayeasmin@bau.edu.bd





## Abstract

Soil organic carbon (OC) is the largest terrestrial C stock and soils' capacity to preserve OC varies with many factors including land use, soil type and depth. We investigated the effect of land use change on particulate organic matter (POM) and mineral-associated organic matter (MOM) in soils. Surface (0-10 cm) and sub-surface (60-70 cm) soil samples were collected from paired-sites (native and cropped lands) of four contrasting soils. Bulk soils were isolated into POM and MOM fractions, which were analysed for mineralogy, OC and nitrogen, isotopic signatures and [14]C content. POMs of surface soils were relatively unaffected by land use change, possibly because of continuous input of crop residues, while corresponding POM in sub-surface lost more OC. In surface soils, oxides-dominated MOM lost more OC than phyllosilicates- and quartz -dominated MOM, which is attributed to diverse OM input and the extent of OC saturation limit of soils. In contrast, oxides-associated fractions were less affected in the sub-surface soils than the other two MOM fractions, possibly due to OC protection via organo–mineral associations. Changed isotopic signature (linked with vegetation) across the fractions suggested that fresh crop residues constituted the bulk of OM in surface soils (supported by greater [14]C). Increased isotopic signatures and lower [14]C in sub-surface MOM fractions suggested the association of more microbially processed, aged OC in oxides-rich fractions than other MOMs. Results reveal that quantity and quality of OC after land use change was influenced by the nature of C input in surface soils and by mineral-organic association in sub-surface soils.





**Keywords**: Soil organic matter, organo-mineral association, organic carbon saturation, microbial decomposition, land use change

## 1   Introduction

Globally, organic carbon (OC) content in the top $0 - 100$ cm soil has been estimated to be between 3,500 and 4,800 Pg C (Lehmann and Kleber, 2015) and nearly a quarter of this amount is present in the top 20 cm of soil (Guo and Gifford, 2002; Jobbagy and Jackson, 2000). The soil OC pool is much greater than other terrestrial pools, i.e., vegetation (420-620 Pg C) and the atmosphere (829 Pg C) (Lehmann and Kleber, 2015). Thus, soils are viewed as a major reservoir and a potential C sink; which could sequester significant quantities of atmospheric $CO_2$. However, whether the soils will act as a sink or source of $CO_2$ is highly dependent on the land use, soil properties (Feller and Beare, 1997) — especially clay minerals, and biophysical factors, including climate (Jobbaggy and Jackson, 2000). According to Six et al. (2002) land use and management are among the most important determinants of soil OC stocks. Land use changes, especially conversion of native grassland and forest to crop land, typically lead to decline in soil OC (Guo and Gifford, 2002), due to a reduction in C input in the soil. Several studies have shown 20 to 50% decline in soil OC when native forest or grassland was converted to cropping (Birch-Thomsen et al., 2007; Bruun et al., 2015; Guo and Gifford, 2002; Luo et al., 2010; McDonagh et al., 2001). The response of soil OC of a native ecosystem to land use conversion depends heavily on the specific vegetation type extent in the native land system and in the system to which the land has been converted (Bruun et al., 2009).





Soil organic matter (OM) is composed of diverse mixtures of OC compounds with differing in
physiochemical properties, degree of stabilisation and turnover rate. So, different OM pools
may show different susceptibility to land use and management. Thus, it appears that the extent
to which land use change influences soil OM dynamics can be best evaluated by separating OM
into fractions (Chenu and Plante, 2006; Jones and Singh, 2014; Sollins et al. 2006, 2009).
Separation of OM pools can be done by different physical fractionation (e.g., particle size,
aggregate, density separation) methods which are effective for separating specific C pools
responsive to land use and management (Collins et al., 1996; Tan et al., 2007). Sequential
density fractionation is able to separate soil OM into labile (light) and stable (heavy) OC pools
differing in structure and function (Sollins et al., 1999; Wander and Traina, 1996) based on
their specific densities (Sollins et al., 2006; von Lu¨tzow et al., 2007). Moreover, this
fractionation method is affirmed to focus the organo-mineral associations in MOM pools
properly which has a prime importance in C turnover dynamics in soils (Basile-Doelsch et al.,

70      2007).

Generally, labile (particulate) OM (POM) is rapidly decomposable (Zimmermann et al., 2007),
hence has a relatively shorter turnover time; and it is often considered more sensitive to land
use conversion than mineral associated OM (MOM) (Gregorich and Janzen, 1996; Leifield and
Kögel-Knabner, 2005; Six et al., 1998). The organo–mineral association usually results in a
reduced biodegradation of OM, due to chemical interactions of OM on reactive mineral surfaces
(Chenu and Plante, 2006). Thus, the MOM pool has a relatively longer turnover time (Kögel-
Knabner et al., 2008) and is presumably less sensitive to land use change. Several studies have
reported a decrease in POM pool as a result of land use change (Conant et al., 2004;





Franzluebbers and Stuedemann, 2002; Six et al., 1998), while others did not find any significant
change in POM resulting from changed land use (Conant et al., 2003; Jastrow, 1996; Leifeld
and Kögel-Knabner, 2005). Moreover, despite the longer turnover time of MOM pool compared
to POM, the former pool may also respond quickly to land use changes (Chenu et al., 2001;
Leifeld and Kögel-Knabner, 2005; Shang and Tissen, 2000). For example, Shang and Tissen
(2000) reported a loss of 19-59% OM from MOM (silt- and clay- sized OM) pool in an Oxisol
(Ferralsol in WRB) due to the land use change from forest to cereal cropping.
There are several characterization methods used for obtaining insights about soil OM pools.
Among them, stable isotope ($\delta^{13}C$ and $\delta^{15}N$) analysis is a powerful tool to assess the source
and/or the degree of microbial transformation of OM in soil (Hobbie and Ouimette, 2009;
Sollins et al., 2009) as well as the dynamics of different soil OM pools with land use change
(Hobley et al., 2017; Leifield and Kögel-Knabner, 2005; Rabbi et al., 2014). Radioactive
isotope of C ($^{14}C$) can provide the information of mean age of C in soils which can indicate the
stability of soil OM (Trumbore, 2009) pools in relation to land use conversion (Schrumpf et al.,
2013) and soil types (Eusterhues et al., 2007).
Although, land use effect on soil OM has been studied extensively, study on the control of soil
types, particularly soil mineralogy in relation to the impact of land use on OC pools is still rare.
Moreover, majority of the studies evaluating the land use change effects on soil OC are limited
to surface ($^{\sim}30$ cm) soil (Bruun et al., 2009; Lorenz and Lal, 2005; Rabbi et al., 2014), as OC
concentration and turnover are usually greater in surface soil (Conant et al., 2001). There is a
growing body of evidence that land use change can also affect the OC dynamics in sub-surface





soil (Don et al., 2011; Poeplau et al., 2011; Wright et al., 2007). Thus, our study was conducted
to evaluate the effects of land use change, from native vegetation to cropping, on the POM and
MOM pools of both surface and sub-surface soils with contrasting mineralogies. This will aid
to identify the sensitivity of the OC pools towards land use conversion varied with depth and
soil types, which ultimately could provide the idea of the best land use practice for the specific
soil to maintain or restore soil health by preserving OC and mitigate global warming.

## 2   Materials and Methods

### 2.1  Site description

We sampled paired sites, i.e. native and cropped, to study the effects of land use conversion on
soil OC dynamics. Four sites were selected in New South Wales, Australia, with each site
representing a different soil type − Ferralsol, Luvisol, Vertisol and Solonetz. The paired sites at
each location represented similar landscape, position, climatic conditions and major soil
characteristics. The native lands were composed of open woodland and cropped sites had been
used for cereal cultivation for over 15 years in all the soil types. Open woodlands had very few
scattered low trees, mainly dominated by Eucalyptus species in association with grass
understorey, and never been grazed. The cultivated cereal was maize, wheat + barley and
sorghum for Ferralsol, Luvisol and Vertisol, respectively. The Solonetz cropped site had been
covered with maize and lucerne. Detailed geological and climatic descriptions for the sites are
given in supplementary information (S) Table S1.





### 2.1 *Soil sampling and general characterisation of bulk soils*


Random bulk soil samples were collected from several spots for the two depths: surface (0–10
cm) and sub-surface (60–70 cm) of each of the paired sites. The sub-surface samples were taken
to represent the absolute mineral soils. The random samples from the corresponding depth were
mixed thoroughly to make one composite sample for each of the individual sites based on the
protocol used in many earlier relevant studies (Kaiser et al., 2010, 2012; Lehmann et al., 2007;
Sleutel et al., 2011; Sollins et al., 2006, 2009). Admittedly, that a sampling strategy with
separate two or three field replications instead of compositing replications at each site would
have been advantageous to find out the spatial variability, but we still believe this sampling
protocol would not limit the capacity of this study to assess land use effects in contrasting soils.
The samples were air dried, ground and passed through a 2 mm sieve. Soil pH and electrical
conductivity (EC) were measured in water using 1:5 soil-to-water ratio. Cation exchange
capacity (CEC) and exchangeable cations were determined by the silver thiourea method
(Chhabra et al., 1975). Particle size analysis was conducted by the pipette method (Gee and
Bauder, 1986). The total Fe and Al concentrations (also Mn and Si) of crystalline pedogenic
oxides were estimated by the dithionite-citrate-bicarbonate (DCB) method (Mehra and Jackson,
1958). Poorly crystalline Fe and Al were quantified by acid ammonium oxalate (pH 3.0)
extraction in the dark (Schwertmann, 1964); and organically complexed Fe and Al were
extracted using Na-pyrophosphate (pH 10.0) (McKeague, 1967). All extracted cations were
analysed using an atomic emission spectrometer (Varian 720-ES). All soil analyses were
performed in triplicate except the particle size analysis where only one replicate was analysed.



## 2.2 Sequential density fractionation


Bulk soil samples from both land use sites and depths were separated into four density fractions
(<1.8 (POM), 1.8–2.2 (1.8DF), 2.2–2.6 (2.2DF) and >2.6 (>2.6DF) g cm$^{-3}$) using the method
adopted from Jones and Singh (2014) and Sollins et al. (2006, 2009) as described in Yeasmin
et al. (2017a, 2017b). The densities were selected with the aim of isolating POM (<1.8 g cm$^{-3}$)
and MOM (>1.8 g cm$^{-3}$) fractions in the studied soils. Briefly, 30 g of air dried soil was weighed
into a 250 ml centrifuge bottle; 125 mL of sodium polytungstate (SPT) solution with a density
of 1.8 g cm$^{-3}$ was added. The contents were shaken for 3 h on a horizontal shaker (300 rpm)
and the suspension was centrifuged for 30 min at 970 $g$. The material floating on the top of SPT
was extracted under suction and SPT was recovered by filtering the supernatant liquid using
0.7 μm glass fiber filter and returned to the same centrifuge tube. The tube was shaken again
for 1 h on a horizontal shaker, centrifuged as described earlier and the floating material aspirated
for second time. The two batches aspirated floating materials (POM: <1.8 g cm$^{-3}$) were
combined and rinsed multiple times with deionised water on a 0.7 μm glass fiber filter to remove
residual SPT until the water EC dropped below 50 μS cm$^{-1}$. The remaining sediment in the
centrifuge bottle from the above fraction step was mixed with 125 mL SPT solution of 2.2 g
cm$^{-3}$ and the whole process was repeated to obtain the 1.8DF (1.8–2.2 g cm$^{-3}$). Similarly, the
next two density fractions (2.2DF and >2.6DF) were obtained using 2.6 g cm$^{-3}$ SPT solution
(2.2DF: supernatant; 2.2-2.6 g cm$^{-3}$ and >2.6DF: Sediment; >2.6 g cm$^{-3}$). The whole
fractionation process replicated twice. After rinsing, all recovered fractions were oven dried at
40°C, hand ground to a fine powder and stored in glass vial for further analyses.





### 2.3 Mineralogical analysis

Mineralogical composition of the bulk soils and density fractions was determined by X-ray diffraction (XRD) analysis using a monochromatic CuKα radiation (GBC MMA diffractometer, Diffraction Technology Pty Ltd, Australia) as described in Yeasmin et al. (2017a). Briefly, the clay fractions (<2 μm) were isolated from bulk soils by the sedimentation method and analysed for both random powder and basally oriented specimen. Density fractions were also analysed as random powder to determine their mineralogical composition. The relative proportion of each mineral in the density fractions expressed semi-quantitatively by integrating mineral peak area using TracesV6 software program (version 6.7.13, GBC Scientific Equipment Pty Ltd, Australia).

### 2.4 Soil organic carbon, nitrogen and stable isotopic ratio analyses

Total C, total N, $\delta^{13}$C and $\delta^{15}$N of the bulk and density fractions were determined by isotopic mass spectroscopy (Thermo Finnigan Delta V). Delta values are expressed in parts per mil (‰) on the Vee Pee Dee Belemnite (VPDB) scale. Duplicate samples were analysed and the precisions for total C, total N, $\delta^{13}$C and $\delta^{15}$N were 0.06–0.5%, 0.01–0.12%, 0.01–0.09‰ and 0.05–0.09‰, respectively. As there were no carbonates in the soil samples, total C values were considered as the OC concentration in the samples.

### 2.5 $^{14}$C analysis

Due to the budget and time limitations, $^{14}$C content was determined using the bulk soils and the 2.2DF MOM fraction of the surface and sub-surface samples from cropped land only. The



selection of the density fraction was based on the result from our previous study (Yeasmin et
al. 2017b), where we found this 2.2DF as the most potential for organo-mineral associations in
the similar soils. For these analyses, samples were pre-treated with 2M HCl at 40°C for at least
24 h until dry to remove possible carbonate contaminants. The treated samples were combusted
for 9 h at 900°C and the resultant $CO_2$ was then reduced to graphite using the Fe/H$_2$ method
(Hua et al., 2001). The graphite was then loaded into an aluminium cathode for accelerator mass
spectrometry (AMS) analysis using the STAR accelerator (Fink et al. 2004). A small portion of
the graphite target was analysed for $\delta^{13}C$ using an elemental analyser/ isotope ratio mass
spectrometer (Vario microcube EA, Elementar, Germany and IsoPrime Isotope Ratio Mass
Spectrometer (IRMS), GV Instruments, UK). The $^{14}C$ content was corrected for isotope
fractionation using measured $\delta^{13}C$, and is reported as percent modern C (pMC) (Stuiver and
Polach, 1977).

### 2.6   Statistical analyses

Pearson's correlation coefficient (r) was calculated to observe the relationships of MOM
associated OC (MOM-OC) with extractable Fe + Al, MOM-OC loss due to land use change
with extractable Fe + Al in soils and aromatic: aliphatic ratio with OC in density fractions using
the software package IBM SPSS 21.0.





## 3 Results

### 3.1 General soil characteristics

Soils from paired sites (native and cropped land) at both depths were non saline (EC 0.09–0.75 dS m$^{-1}$) and acidic in reaction (pH ≤ 6.8), except for the sub-surface soils of the Vertisol, which were slightly alkaline (pH 7.7–7.8) (Table 1). The CEC of each soil did not vary under different land uses but increased with depth for the Luvisol, Vertisol (native) and Solonetz soils and decreased slightly with depth in the Ferralsol. In general, the Vertisol had the highest CEC (200–231 mmol$_c$ kg$^{-1}$), followed by the Luvisol (182–222 mmol$_c$ kg$^{-1}$), the Ferralsol (69–96 mmol$_c$ kg$^{-1}$) and the Solonetz (36–50 mmol$_c$ kg$^{-1}$).

The textures of the studied soils were different; the surface soil of the Vertisol had the highest clay content (61%), followed by the Ferralsol (34%), Luvisol (31%), and the Solonetz had the lowest content (5%) regardless of land use. The clay content almost doubled in the sub-surface samples of the Ferralsol and Luvisol, but there was no noticeable change in the Vertisol (Table 1). In the Solonetz, the clay content increased about six-fold and ten-fold in the sub-surface soil of the native and cropped sites, respectively.

The extractable Fe and Al values were substantially different for the soils at corresponding depth and land use (Table S2). The highest amount of total (DCB-extracted) Fe and Al was found in the Ferralsol, followed by the Luvisol, Vertisol and Solonetz in regardless of the land use and depth. Poorly crystalline Fe (Fe$_{ox}$) and Al (Al$_{ox}$) concentrations were also highest in the Ferralsol (except for Fe in the native surface Vertisol) followed by the Vertisol, Luvisol and Soloentz. The Fe$_{ox}$:Fe$_{DCB}$ ratio expresses the fraction of poorly crystalline Fe of the total Fe





concentration: Vertisol had the highest $Fe_{ox}:Fe_{DCB}$ ratio, ranging from 0.27–0.53, followed by
the Solonetz (0.04–0.31) and Luvisol (0.09–0.19). The Ferralsol had the lowest $Fe_{ox}: Fe_{DCB}$
ratio (0.03–0.09), suggesting the presence of greater proportions of crystalline Fe (e.g., goethite
and hematite) in this soil (Table S2). A slight decrease in the $Fe_{ox}:Fe_{DCB}$ ratio in the sub-surface
depth was noticed in all soils. This was due to the relatively faster crystallisation of amorphous
Fe in the sub-surface soils compared to that in surface soils, because of their lower OM contents
(Schwertmann and Cornell 1991). Pyrophosphate extractable Fe ($Fe_{Na-py}$) and Al ($Al_{Na-py}$)
ranged from 0.05–9.0 and 0.04–3.7 g $kg^{-1}$, respectively, and the order of the abundance was:
Ferralsol > Luvisol > Solonetz > Vertisol. Organically complexed (Na-py extractable) Fe and
Al concentrations decreased noticeably in sub-surface soils, compared with surface soils, for
both types of land use. There was no clear effect of land use change on pyrophosphate
extractable Fe and Al.
*3.2   Mineralogy of density fractions*
The oriented and random powder of the clay fractions of the bulk soils showed contrasting
mineralogy for the studied soils (Figs. S1, S2, S3i and S4i). The Ferralsol clay fraction showed
the dominant presence of goethite, hematite and gibbsite. Kaolinite and illite were dominant in
the clay fraction of the Luvisol, and present in moderate amounts in the Solonetz. The Vertisol
was dominated by smectite, with kaolinite present in moderate amount. Feldspars and quartz
were also identified in the clay fraction of Luvisol, Vertisol and Solonetz. The mineralogy of
the paired soils was similar (Figs. S1, S2, S3i and S4i), but the overall mineral presence became
slightly prominent in the XRD patterns of the sub-surface soils (Figs. S2 and S4i).





The POM fraction of all the surface soils was dominated by OM, as demonstrated by the
presence of the broad hump in the ~3.55 to 4.45° (d-spacing) region of the XRD patterns (Fig.
S3ii). Trace to small amounts of kaolinite, hematite and gibbsite were present in the Ferralsol,
kaolinite and illite in the Luvisol and Solonetz, and kaolinite and smectite in the Vertisol (Table
2). The proportion of these minerals then increased with increasing density. Greater proportion
of hematite, goethite and gibbsite was observed in the three MOM fractions (1.8DF, 2.2DF and
>2.6DF) of the Ferralsol. A trace to moderate presence of kaolinite was also observed in these
Ferralsol fractions. On the other hand, 1.8DF and 2.2DF of the other soils were dominated by
phyllosilicates; kaolinite and illite in the Luvisol, kaolinite and smectite in the Vertisol and
kaolinite only in the Solonetz. These fractions also showed the presence of feldspars and quartz,
and the proportion of these minerals increased in the heaviest fraction (>2.6DF) while the
phyllosilicate proportion decreased. Similar to the bulk soils, the mineralogy of the density
fractions of the soils did not vary with land uses, but did show some minor differences between
the surface and sub-surface soils (Table 2) For example, there was a slight increase in the
proportion of kaolinite, hematite and anatase in the Ferralsol fractions, an increase in the
proportion of phyllosilicates in the other three soils, and an increase in some oxide minerals in
the Vertisol (hematite and anatase) and Solonetz (goethite and anatase).
It is evident that three different mineral phases were dominant in the MOM fractions (>1.8 g
cm$^{-3}$) of the four soils regardless of the land use and depth: Fe and Al oxides in the Ferralsol
fractions (1.8DF, 2.2DF and >2.6DF); phyllosilicates in the 1.8DF and >2.2DF; and feldspars
and quartz (primary minerals) in >2.6DF of the Luvisol, Vertisol and Solonetz. These three





MOM fractions are referred to as oxide-OM, phyllosilicate-OM and quartz-OM, respectively,
throughout the manuscript.
*3.3   Organic carbon and nitrogen contents of bulk soil and density fractions*
Bulk samples of the Ferralsol had the greatest OC concentration (12–63 g kg$^{-1}$) amongst the
four soils regardless of the land use and depth (Table 3). The OC concentration in all bulk sub-
surface samples was much lower than their corresponding surface samples. There was a
decrease in OC concentration with land use change from native vegetation to cropping in all
soils. The N concentration (0.3–6 g kg$^{-1}$) followed a trend similar to that of OC concentration
in all soils. The C:N ratio in the soils ranged from 7–14, with generally similar values for the
surface soils and a slight variation among sub-surface soils.
The POM fraction of the Solonetz and Luvisol surface samples under native vegetation
(remnant woodland) contributed 33–73% and 21–49% to the total OC and N, respectively (Fig.
1a1, b1). In the Ferralsol (tropical forest) and Vertisol (native grasses) surface soils, the
contribution of POM to total OC was only 7–10% and thus a relatively much greater MOM
contribution to total OC than in the other two soils (Fig. 1a1). The OC concentration in the
POM fraction of the native surface soils ranged between 240 and 355 g kg$^{-1}$ (Table 3), with the
highest concentration in the Ferralsol compared to the other three soils. Among the MOM
fractions (>1.8 g cm$^{-3}$), the oxide-OM (1.8DF + 2.2 DF + >2.6DF) fractions of the Ferralsol
had also greater OC concentrations (245–351 g kg$^{-1}$) than the phyllosilicate-OM (1.8DF + 2.2
DF) fractions of the Solonetz (125–138 g kg$^{-1}$), Luvisol (111–124 g kg$^{-1}$) and Vertisol (94–101
g kg$^{-1}$) soils, irrespective of the type of land use (Table 3). The quartz-OM fraction (>2.6DF)





of these latter three soils had the smallest OC concentration range, at 0.4–6 g kg$^{-1}$. The
distribution of N concentration among the MOM fractions also followed the same pattern as
OC in the soils: oxide-OM > phyllosilicate-OM > quartz-OM fractions. The C:N ratio varied
widely in the density fractions of all soils, between 3 and 43, and showed a decreasing pattern
with increasing density with few exceptions in both land uses (Table 3). Overall, the C:N ratio
was greater in the oxide-OM fractions, followed by the phyllosilicate-OM and then the quartz-
OM fractions.
There was a notable difference in the OC and N concentrations with land use change of each
soil at corresponding depth (Table 3). After clearing the native vegetation and shifting to annual
cropping, the overall POM still accounted for a major proportion (21–40%) of the total OC in
the Luvisol (forage and cereal), Solonetz (improved pasture) and Vertisol (cereal) as compared
to the Ferralsol OC (4%) (Fig. 1). This was also reflected in the OC concentration change (Fig.
2a1); with the former three soils lost only 5–8% of the POM-OC, whereas this loss was 20% in
the Ferralsol POM fraction. The OC concentration loss with land use change increased to 50%
in the oxide-OM fractions of the Ferralsol. The loss was lower in the MOM fractions of the
other three soils, with a maximum of 30% loss in the phyllosilicate-OM fractions and 29% lost
from the quartz-OM fraction. Nitrogen concentration was depleted most from the
phyllosilicate-OM fractions of the Vertisol (54%) followed by the oxide-OM fractions of the
Ferralsol (49%), Solonetz (31%) and Luvisol (24%) phyllosilicate-OM fractions (Fig. 2b1).
Surprisingly, the quartz-OM fractions either lost very little N (1%) or N concentration was
increased (28%) with the land use change to cropping. This resulted in a decreased C:N ratio in
the quartz-OM fraction of the Luvisol, Vertisol and Solonetz (Fig. 2c1). The C:N ratios changed



less or increased in the POM and phyllosilicate-OM fractions of these three soils. The Ferralsol
fractions depleted relatively more in C:N ratio, except in the >2.6DF fraction.
Considering the land uses and soil depths, variation was observed in fraction recovery (Fig. 1),
particularly in the Luvisol and Solonetz samples (82-100%), which might cause overall
increased proportion in the fractions where recovery was higher. Considering this fact, major
contribution of the MOM fractions to the total OC was still evident (Fig. 1a2), mostly in all
sub-surface soils under both land uses, compared with surface soils (Fig. 1a1). The proportion
of N to the total N in fractions also followed a similar pattern to that for OC (Fig. 1b2), with a
noticeably greater contribution of the phyllosilicate-OM to the total N compared to the oxide-
OM and quartz-OM fractions. Overall, OC (1–409 g kg$^{-1}$) and N (0.1–15 g kg$^{-1}$) concentrations
decreased remarkably with depth in all soils and showed similar trends in the density fractions
(POM > oxide-OM > phyllosilicate-OM > quartz-OM) with those of the surface soils (Table
3). However, an increase in OC concentration in the POM fraction of the Ferralsol and Luvisol
was noticed in sub-surface under both type of land uses. The C:N ratio became wider (3–90)
and the increasing trend of the ratio in the density fractions of all soils was more prominent in
the sub-surface soils than in the surface samples (Table 3).
With land use change, OC concentration reduced by 7–27% in the POM fraction of all sub-
surface soils, in the order: Solonetz > Vertisol >Ferralsol > Luvisol (Fig. 2a2). In sub-surface
MOM fractions, the OC concentration decreased the least in the oxide-OM (7–18%) of the
Ferralsol, compared to 7–48% loss in the phyllosilicate-OM fractions (Solonetz > Luvisol >
Vertisol) and 5–25% loss in the quartz-OM fraction. The N concentration (Fig. 2b2) reduced



most from both the POM fractions of all sub-surface soils and the oxide-OM of the Ferralsol.
The C:N ratio decreased (3–23%) in the POM fractions of all soils in sub-surface depth (Fig.
2c2). The oxide-OM fractions ratio showed less depletion/increase (-3 to 42%). The ratio
mostly decreased in phyllosilicate-OM fractions (-57 to 3%) in the order Solonetz > Luvisol >
Vertisol, and this reduction continued in the quartz-OM fractions (-13 to -57%) of sub-surface
samples.
*3.4   Stable isotopic signatures of bulk soil and density fractions*
The Ferralsol bulk soils had lower $\delta^{13}C$ (-25.9 to -22.3‰) and higher $\delta^{15}N$ (6.2–9.7‰) values,
as compared with the other three soils, irrespective of land use and depth (Table 3).
In general, both isotopic values were lowest in the POM fraction of all surface soils under both
land uses, with $\delta^{13}C$ values ranging from -27.1 to -21.1‰ and $\delta^{15}N$ ranging from 1.7 to 8.1‰
(Table 3). The $\delta^{13}C$ and $\delta^{15}N$ values then increased (-27.2 to -20.4‰ for $\delta^{13}C$; 1.7-10.8 ‰ for
$\delta^{15}N$) with the density in all surface soils for both land uses. The increment continued
throughout all MOM fractions of the Ferralsol, but increased up to 2.2DF for the Luvisol,
Vertisol and Solonetz and then decreased in the >2.6DF. Oxide-OM fractions had overall lower
$\delta^{13}C$ and $\delta^{15}N$ values, compared with the phyllosilicate-OM and quartz-OM fractions. After
land use change, $\delta^{13}C$ value change ($\Delta\delta^{13}C$) across the density fractions showed an enrichment
of $^{13}C$ by 0.4–4.8‰, with a pattern increasing with density in the Ferralsol, which was different
to other three soils (Fig. 3a1). The Vertisol fractions had enrichment of $^{13}C$ but to a smaller
extent (0.9–1.6‰), while the Luvisol fractions showed an overall depletion of $^{13}C$ (up to -
1.5‰). The POM fraction had a large $^{13}C$ enrichment (4.6‰) in the Solonetz fractions but the



enrichment then decreased with density (up to 1.5‰). The $\delta^{15}N$ value change ($\Delta\delta^{15}N$) showed
a depletion of $^{15}N$ in all fractions of the Ferralsol (up to -3.8‰), Luvisol (up to -2‰) and
Solonetz (up to -0.6‰, except the >2.6DF), whereas there was an enrichment of $^{15}N$ (up to
2.3‰) in the Vertisol fractions (Fig. 3b1).
The $\delta^{13}C$ and $\delta^{15}N$ values in the density fractions of sub-surface soils followed trends similar
to surface soils; i.e., increased with increasing density and lower values in the Ferralsol fractions
than in the other soils (Table 3). However, both isotopic values increased (-25.9 to -11.7 ‰
$\delta^{13}C$, 3.4–14.4 ‰ $\delta^{15}N$) in the density fractions of all soils with depth under both land uses.
Prominent enrichment of isotopic values in sub-surface density fractions also exhibited with
land use change (Fig. 3a2). In general, $\delta^{13}C$ value enrichment was lower in the POM fractions
(up to 2.2‰); it then increased (up to 3.8‰) in the MOM fractions of all soils in the order:
oxide-OM > phyllosilicate-OM (Vertisol > Luvisol > Solonetz) > quartz-OM (Fig. 3a2). The
$\delta^{15}N$ change showed an overall consistent enrichment (up to 4.5 ‰) with increasing density in
all soils after land use conversion (Fig. 3b2).
*3.5   $^{14}C$ content*
The $^{14}C$ content value in all the surface bulk soils from cropped site and their 2.2DF fractions
was >100 pMC (101.4-108.1), except in the Vertisol bulk soils (99.7 pMC), indicating the
presence of modern OC (Table 3). Modern OC means the OC bears a signature of post-bomb
(post-1950) $^{14}C$ content (higher than 100 pMC) (Hua et al., 2013). In the sub-surface cropped
soils, $^{14}C$ content ranged from 60.2 - 81.3 pMC in the bulk soils and 62.7 - 107.8 pMC in the
2.2DF fractions, suggesting relatively older OC ($^{14}C$ content was lower than 100 pMC)





compared to their corresponding surface soils. The $^{14}$C content of the sub-surface bulk soils was
in the order of: Ferralsol < Vertisol < Solonetz < Luvisol and it was in the order: Vertisol <
Ferralsol < Luvisol < Solonetz in the 2.2DF fractions (Table 3). The $^{14}$C content of 2.2DF
fractions of cropped soils was negatively correlated with the corresponding $\delta^{13}$C ($R^2$ = -0.41)
and $\delta^{15}$N ($R^2$ = -0.24) (Fig. 4).
**4    Discussion**
*4.1    General trends of organic carbon throughout the density fractions of surface*
*and sub-surface soils under native and cropped land use*
Narrower C:N ratio and larger $\delta^{13}$C and $\delta^{15}$N values in the MOM fractions, compared to the
POM fractions, in all the soils (Table 3) suggest a more advanced stage of decomposition of
OM in the MOM fractions (Baldock et al., 1992; John et al., 2005). Generally, the C:N ratio in
soils decreases with depth (Rumpel and Kögel-Knabner, 2011), which is ascribed to more
microbially processed OC (Boström et al., 2007). The C:N ratio in the MOM fractions of the
sub-surface soils of this study also showed an overall decrease (or slight increase); however, an
increase in the C:N ratio in the POM fractions of all soils was observed (Table 3). This trend
was more evident in the Ferralsol. The increased C:N ratio in the sub-surface POM fractions
could be due to the lack of microbial processing (Schrumpf et al., 2013). However, the increased
stable isotopic values ($\delta^{13}$C and $\delta^{15}$N) do not support this assertion (Table 3). Increased C:N
ratios in the POM fractions could be related to the composition of the below-ground OM source.
It has been reported that fine roots in sub-surface soils have lower N contents than the surface
soil, which can result in a large C:N ratio (Ugawa et al., 2010). The increased C:N could also



be due to the presence of charred organic material (Rumpel and Kögel-Knabner, 2011),
however we did not specifically measured the charcoal C concentration in the samples.
*4.2 Effect of land use change on surface soil organic carbon: influence of*

*vegetation type*

The POM fractions of surface mineral soils are known to be more sensitive to land use change
than the MOM fractions, due to their relatively fast turnover rates and close link to litter input
(Bird et al., 2007; Ellerbrock and Gerke, 2013; Golchin et al., 1995; Haynes, 2005). The
changes in the proportion (Fig. 1a1) and concentration of OC (Fig. 2a1) in the surface soil
fractions in this study contradict with these previously reported results. In our study, the MOM
fractions of the surface soils lost relatively more OC with land use change than the POM
fractions. Input of organic residues from annual crops probably compensated the decomposed
POM, which resulted in a small loss of POM-OC after converting native land to cropping land.
Ontl et al. (2015) reported an increase in unprotected POM-OC pool after monitoring OM in
soil with different cropping systems; i.e., grass and cereals for three years. They attributed the
gain of this pool to the least decomposed and most recent deposited OM from crop residues.
The pattern of OC concentration in the MOM fractions (Table 3) of the four soils (oxide-OM >
phyllosilicates-OM > quartz-OM) indicates the role of minerals, especially the surface
reactivity of mineral phases, in OC stabilisation (Kögel-Knabner and Kleber, 2011). It has been
extensively reported that soil with Fe (hydro)oxides and poorly crystalline Al silicates have a
greater potential to protect more OC than the soils dominated by phyllosilicates and primary





minerals (Eusterhues et al., 2005; Kögel-Knabner et al., 2008; Kögel-Knabner and Kleber,
2011). Significant positive correlations of MOM-OC to total (DCB) and poorly crystalline
(oxalate) Fe and Al oxides support this hypothesis (Fig. S5). With larger specific surface area
(SSA) and hydroxyl surface groups, these minerals are able to form stronger bonds via ligand
exchange to associate with OC, as compared to the weaker cation bridge on phyllosilicates (Gu
et al., 1995; Kaiser and Zech, 2000; Yeasmin et al., 2014). Consequently, OC adsorbed to oxide
mineral surfaces is expected to be more protected from decomposition than the OC associated
with permanent charged phyllosilicates (Yeasmin et al., 2014).
Organic C loading based on the specific surface area of minerals is also important; as this can
control the stability of the organo–mineral associations (Kaiser and Guggenberger, 2003). The
capacity of soils to stabilise OC can be restricted by their OC saturation limit (Hassink, 1997).
When a soil approaches its OC saturation, OC stability declines, probably to the changes in type
and strength of OC–mineral interactions with increasing OC loading (Kleber et al., 2007;
Sollins et al., 2009). As surface loading increases beyond a certain threshold, the new OC might
not form direct bonds to mineral surfaces but may instead form organic multilayers through
hydrophobic interactions or polyvalent cation bridge between organic ligands of already sorbed
and new organic molecules (Kaiser and Guggenberger, 2003). In the zonal model for OM
stabilisation, Kleber et al. (2007) proposed the occurrence of greater abundant weaker organic–
organic interactions than stronger organo–mineral interactions in soils with increasing OC
loadings. Therefore, the weakly sorbed OC may be relatively more sensitive to land use change
being more readily available for microbial decomposition than the directly bonded and strongly
sorbed OC (Krull et al., 2003). The greater loss of OC and N from MOM fractions of the studied





soils (Fig. 2a1,b1) is consistent with the OC saturation concept. The soils in the study,
particularly Ferralsol, possibly received large organic residues input when under native
vegetation, which subsequently added more OC to the soil (Table 3). Therefore, the MOM
fractions may have already attained their OC saturation point and a significant proportion of
OC was associated by forming weak organic–organic multilayers under native vegetation land
use (Kleber et al., 2007). Although we did not measure the saturation point of our soil, however,
if we compare the data of our soils with the OC saturation curve derived by Fujisaki et al.
(2018), it showed that the soil had already reached the saturation point. Since the soil had OC
greater than its saturation point, the land use change to agricultural cropping resulted in the
microbial mineralisation of the OC that was retained by weak association with soil minerals.
The change in OC and N concentrations in the density fractions of the surface soils was more
influenced by the vegetation type rather than the soil mineralogy. The significant negative
correlations (r = -0.97) of MOM-OC depletion with total and poorly crystalline Fe and Al oxide
concentrations (Fig. 5a) in the cropped surface soils indicate a weaker control of soil minerals
on OC loss due to the effect of land use change. This perception is also supported by the change
in $\delta^{13}$C and $\delta^{15}$N values. Generally, changes in stable C isotopic composition ($^{13}$C/$^{12}$C) of soil
occur for two reasons (Werth and Kuzyakov, 2010): (i) preferential stabilisation of substrates
with $^{12}$C (e.g., lipids, phenols, lignin) or $^{13}$C (e.g., cellulose, amino acids, hemicellulose) and/or
(ii) stabilisation of microbial products that are enriched in $^{13}$C after one or more microbial
utilisation cycles (because of release of $CO_2$ with $^{12}$C) (Guina and Kuzyakov, 2014; Sollins et
al., 2006; von Lützow et al., 2006). The second mechanism (microbial utitlisation) causes
greater $^{13}$C enrichment than the first mechanism (Guina and Kuzyakov, 2014); this $^{13}$C





increment also shows an increased pattern with the density. However, the trend of $\delta^{13}$C change
(either decrease or increase) among the POM and MOM fractions were marginal and/
inconsistent in all soils except the Ferralsol (Fig. 3a1). The might suggests accumulation of less
decomposed OC under high OC input in surface soils (Margenot et al., 2015). The overall high
$^{14}$C content in the bulk soils and MOM (2.2DF) fraction also supports the presence of young C
in surface soils (Table 3). Therefore, we hypothesise that the change in stable isotopic values
reflects the change in the isotopic signature of the OM source; i.e., the vegetation type rather
than the preferential stabilisation or microbial utilisation of OC. Although we did not estimate
the share of different types of vegetation C in OC pool after land use change, we could relate
the isotopic signature variations with the information of the existing vegetation types (C3 or
C4) in the paired sites.  It is known that C3 plant species have a lower $\delta^{13}$C value (~ -23 to -
40‰) than C4 plant species (~-9 to -19‰). The Ferralsol native open woodland site was
perhaps covered by both C3 and C4 plants. When the land use shifted to a C4 maize crop, the
$\delta^{13}$C value increased because of the dominance of the input of OM from C4 plants. Mixed plant
species (C3 and C4) supposedly also existed on the Luvisol and Vertisol native sites
(woodland), and after clearing and growing C3 cereals, such as wheat and barley, might cause
an overall decrease in $\delta^{13}$C of all the fractions of these soils.
*4.3  Effect of land uses on sub-surface soil organic carbon: influence of soil*

*mineralogy*

A greater potential of sub-surface soil for OC preservation has been explicitly proven by
isolating older C in the separated OM pools of sub-soils than its bulk OM (Helfrich et al., 2007;





Paul et al., 2001; Rumpel et al., 2008) and surface soil OM (Bruun et al., 2008; Hobley et al.,
2017). Prolonged protection of OC in the sub-surface soil has been attributed to stronger
organo–mineral interactions with smaller OM loading (Kögel-Knabner et al., 2008; Schrumpf
et al., 2013). However, Fontaine et al. (2007) reported reduced input of OM; i.e., energy
limitation for microbes in the sub-surface soil, as the reason for slow decomposition of OM,
rather than strong organo–mineral association. They showed that supplying of fresh plant-
derived C stimulated the decomposition of old OM in the sub-surface soil (60–80 cm). Linking
these two theories, we speculate that there might be a threshold point of OC input/energy supply
at which the microbial activity becomes minimum, and that could also be variable among soils,
land uses and climatic conditions. This assumption is in line with the trend of OC loss from
both POM and MOM fractions in the sub-surface soils of this study (Fig. 2a2). Overall POM-
OC losses were similar in all soils, while the loss of MOM-OC was greater in the Luvisol,
Vertisol and Solonetz than in the Ferralsol (Fig. 2a2). Reduced OM input (compared to the
surface layer) was perhaps still greater than the previously mentioned threshold point that led
to continued microbial activity in the sub-surface soils. Under low OC input conditions,
microbes probably first utilised the most easily available POM fractions and then mineralised
relatively weakly bound OC to mineral surfaces in the MOM fractions. Greater reduction in the
POM-OC of the sub-surface soils than the surface soils (Fig. 2a1,a2), and subsequent variable
MOM-OC losses among the soils, i.e. greater losses in the phyllosilicate-OM and quartz-OM
fractions of the Luvisol, Vertisol and Solonetz than in the oxide-OM of the Ferralsol (Fig. 2A2),
support our hypothesis. Desorption of the relatively weakly bound OC from phyllosilicates and





quartz was probably enhanced by mechanical disturbance during tillage in cropped lands, which
caused greater OC loss with the change in land use.
In general, variable losses of OC with land use conversion indicate the influence of soil
mineralogy. This is supported by a significant positive correlation between change in the MOM-
OC and poorly crystalline Fe and Al oxides in the cropped soils (Fig. 5b). The data indicate
smaller losses of OC with increased concentration of poorly crystalline Fe and Al oxides in the
sub-surface soils. Kasier and Guggenberger (2000) also emphasised the importance of Fe
oxides in providing surface area for OC sorption in sub-surface soils. Kögel-Knabner et al.
(2008) also suggested Fe oxides as more important sorbents than phyllosilicates, for the
formation of organo–mineral associations in sub-surface soils. Parafitt et al. (1997) also
reported greater OC stabilisation in Fe oxides and poorly crystalline Al silicates rich Andisol
and Inceptisol, which was resistant to land use change from pasture to cropping. In this present
study, relatively greater radiocarbon age of the oxides-MOM fraction of the Ferralsol compared
to age (average) of the phyllosilicates-OM fractions of the other three soils establish the
potential of Fe and Al oxides in long-term OC stabilisation in soil.
The distinct increasing enrichment trend of both $^{13}$C and $^{15}$N in the density fractions of all sub-
surface soils (Fig. 3a2,b2) demonstrated the presence of more microbially processed OC and
supports the hypothesis of OC stabilisation by association with minerals (Gunina and
Kuzyakov, 2014) in sub-surface soils. However, the increased C:N ratio in the Ferralsol
fractions (as opposed to the other soils) did not support the microbial processing hypothesis
(Fig. 2c2). This could happen for several reasons, such as (i) C-rich OM input in the cropped





land which leached down to the sub-soil, (ii) enrichment of microbial-derived C, such as sugar
(Spielvogel et al., 2008), and (iii) selective decomposition of N rich OM (Yeasmin et al., 2014).
Greater loss of N than of OC in the Ferralsol fractions supports the selective utilisation of N-
rich OM in the Ferralsol hypothesis (Fig. 2a2,b2). Nitrogen depletion pattern in the MOM
fractions of the studied soils, which was greater in the Ferralsol than the other soils (Fig. 2b2),
might suggest a higher affinity of phyllosilicates and quartz minerals towards N-rich OC
compounds (Jagadamma et al., 2010; Mikutta et al., 2009); this is consistent with the overall
higher depletion of C:N ratios in the Luvisol, Vertisol and Solonetz soils (Fig. 2c2).
**5    Conclusions**
Our results show that the land use change impacts the OC in both surface and sub-surface soils.
The MOM fractions are not necessarily resistant to land use conversion, particularly in the
surface soils. Inconsistent changes in $\delta^{13}$C and $\delta^{15}$N values (close to crop residue), a decreasing
trend in the OM decomposition index and a greater $^{14}$C content highlight the vegetation effect
on the OC change through fresh OM supply in the surface soils. Under different OC loading
conditions in surface soil, POM was less sensitive due to continuous input of agricultural crop
residues. Among the MOM fractions, oxide-OM fractions of the Ferralsol lost more OC than
did the phyllosilicate-OM and quartz-OM fractions of the Luvisol, Vertisol and Solonetz after
land use changes. The variable OC loss in the MOM fractions can be attributed to greater OM
supply in the Ferralsol and possible OC saturation on the mineral surface. In sub-surface soils
with limited OC supply, the POM fraction was more sensitive to land use change than in the
surface soil. In the MOM fractions of the sub-surface soils, oxides-OM fractions of the Ferralsol





preserved more OC than did the phyllosilicate-OM and quartz-OM fractions of the Luvisol,
Vertisol and Solonetz, after land use changes. The OC accumulated in the MOM fractions of
sub-surface soils was highly microbially processed (enriched in $\delta^{13}$C and $\delta^{15}$N) and relatively
older (lower $^{14}$C content) in nature.
Organic matter concentration in both the POM and MOM fractions of the surface soils is highly
sensitive to land use change. The association of OM with minerals is more relevant in
preserving soil OC and controlling the impact of land use change in the sub-surface soils. Sub-
surface soils can act as a potential C sink for long-term C storage, together with improved land
use and vegetation management systems.
**Data availability**
All data have been included in the supporting information of this article.
**Author contribution**
Yeasmin, S. planned and conducted the experiment, performed data analysis and critical
interpretation of data and wrote the manuscript. Singh, B. helped plan the experiment,
supervised the laboratory work and was involved in interpretation of the results and reviewed
the manuscript. Johnston, C.T. supervised DRIFT data analysis and reviewed the manuscript.
Sparks, D.L. reviwed and edited the manuscript. Quan, H. performed radiocarbon analysis, $^{14}$C
(pMC) calculation and helped to interpret the results and reiewed the manuscript.





**Competing interests**
The authors declare that they have no conflict of interest.
**Acknowledgements**
The corresponding author acknowledges the financial support of the International Postgraduate
Research Scholarships and Postgraduate Research Support Scheme of the University of
Sydney, NSW, Australia. The authors express gratitude to Australian Institute of Nuclear
Science and Engineering for providing a research grant (ALNGRA15536) for AMS [14]C
analysis. We also thank Dr Claudia Keitel of the Centre for Carbon, Water and Food, The
University of Sydney, and Dr Elizabeth Carter and Dr Joonsup Lee of the Vibrational
Spectroscopy Facility at The University of Sydney, for technical and analytical supports in mass
spectroscopic analysis and DRIFT analysis, respectively. All data have been included in the
supporting information of this article.

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



Methods Phys Res Section B: Beam Interactions with Materials and Atoms, 223, 109-115,
https://doi.org/10.1016/j.nimb.2004.04.025, 2004.
Fontaine, S., Barot, S., Barré, P., Bdioui, N., Mary, B., and Rumpel, C.: Stability of organic carbon in
sub-surface soil layers controlled by fresh carbon supply. Nature, 450, 277-281,
https://doi.org/10.1038/nature06275, 2007.
Franzluebbers, A.J., and Stuedemann, J.A.: Particulate and non-particulate fractions of soil organic
carbon under pastures in the Southern Piedmont USA. Environ. Pollut., 116, S53-S62,
https://doi.org/10.1016/S0269-7491(01)00247-0, 2002.
Fujisaki, K., Chapuis-Lardy, L., Albrecht, A., Razafimbelo, T., Chotte, J.L., and Chevallier, T.: Data
synthesis of carbon distribution in particle size fractions of tropical soils: Implications for soil
carbon storage potential in croplands. Geoderma, 313: 41-51,
https://doi.org/10.1016/j.geoderma.2017.10.010, 2018.
Gee, G.W., and Bauder, J.W.: Particle-size analysis, in: Methods of Soil Analysis: Part 1-Physical and
Mineralogical Methods, 2nd edn, edited by: Klute, A., Soil Sci Soc Am J, American Society of
Agronomy, Madison, WI, USA, 383-411, 1986.
Golchin, A., Oades, J.M., Skjemstad, J.O., and Clarke, P.: Structural and dynamic properties of soil
organic-matter as reflected by $^{13}$C natural-abundance, pyrolysis mass-spectrometry and solid-state
$^{13}$C NMR-spectroscopy in density fractions of an oxisol under forest and pasture. Soil Res., 33, 59-
76, https://doi.org/10.1071/SR9950059, 1995.





Gregorich, E.G., and Janzen, H.H.: Storage and soil carbon in the light fraction and macroorganic matter.
in: Structure and Organic Matter Storage in Agricultural Soils, edited by: Carter, M.R., and Stewart,
B.A., Advances in Soil Science. CRC press, Boca Raton, 167-190, 1996.
Gu, B., Schmitt, J., Chen, Z., Liang, L., and McCarthy, J.F.: Adsorption and desorption of different
organic matter fractions on iron oxide. Geochim. Cosmochim. Acta, 59, 219-229,
https://doi.org/10.1016/0016-7037(94)00282-Q, 1995.
Gunina, A., and Kuzyakov, Y.: Pathways of litter C by formation of aggregates and SOM density
fractions: Implications from $^{13}$C natural abundance. Soil Biol. Biochem., 71, 95-104,
https://doi.org/10.1016/j.soilbio.2014.01.011, 2014.
Guo, L.B., and Gifford, R.M.: Soil carbon stocks and land use change: a meta analysis. Global Change
Biol., 8, 345-360, https://doi.org/10.1046/j.1354-1013.2002.00486.x, 2002.
Hassink, J.: The capacity of soils to preserve organic C and N by their association with clay and silt
particles. Plant Soil, 191, 77-87, https://doi.org/10.1023/A:1004213929699, 1997.
Haynes, R.J.: Labile organic matter fractions as central components of the quality of agricultural soils:
an overview. Adv. Agron., 85, 221-268, 2005.
Helfrich, M., Flessa, H., Mikutta, R., Dreves, A., and Ludwig, B.: Comparison of chemical fractionation
methods for isolating stable soil organic carbon pools. Eur. J. Soil Sci., 58, 1316-1329,
https://doi.org/10.1111/j.1365-2389.2007.00926.x, 2007.



Hobbie, E.A., Ouimette, A.P.: Controls of nitrogen isotope patterns in soil profiles. Biogeochemistry,

95, 355-371, https://doi:10.1007/s10533-009-9328-6, 2009.

Hobley, E., Baldock, J., Hua, Q., and Wilson, B.: Land-use contrasts reveal instability of subsoil organic

carbon. Global Change Biol., 23, 955-965, https://doi.org/10.1111/gcb.13379, 2017.

Hua, Q., Barbetti, M., and Rakowski, A.Z.: Atmospheric radiocarbon for the period 1950-2010.

Radiocarbon, 55, 2059-2072, https://doi.org/10.2458/azu_js_rc.v55i2.16177, 2013.

Hua, Q., Jacobsen, G.E., Zoppi, U., Lawson, E.M., Williams, A.A., Smith, A.M., and McGann, M.J.:

Progress in radiocarbon target preparation at the ANTARES AMS Centre. Radiocarbon, 43, 275-

282, https://doi.org/10.1017/S003382220003811X, 2001.

Jagadamma, S., Lal, R., Ussiri, D.A., Trumbore, S.E., and Mestelan, S.: Evaluation of structural

chemistry and isotopic signatures of refractory soil organic carbon fraction isolated by wet oxidation

methods. Biogeochemistry, 98, 29-44, https://doi.org/10.1007/s10533-009-9374-0, 2010.

Jastrow, J.D.: Soil aggregate formation and the accrual of particulate and mineral-associated organic

matter. Soil Biol. Biochem., 28, 665-676, https://doi.org/10.1016/0038-0717(95)00159-X, 1996.

Jobbaggy, E.G., and Jackson, R.B.: The vertical distribution of soil organic carbon and its relation to

climate    and    vegetation.    Ecol.    Appl.,    10,    423-436,    https://doi.org/10.1890/1051-

0761(2000)010[0423:TVDOSO]2.0.CO;2, 2000.



John, B., Yamashita, T., Ludwig, B., and Flessa, H.: Storage of organic carbon in aggregate and density

fractions of silty soils under different types of land use. Geoderma, 128, 63-79,

https://doi.org/10.1016/j.geoderma.2004.12.013, 2005.

Jones, E., and Singh, B.: Organo-mineral interactions in contrasting soils under natural vegetation.

Front. Environ. Sci., 2, 2. https://doi.org/10.3389/fenvs.2014.00002, 2014.

Kaiser, K., and Guggenberger, G.: Mineral surfaces and soil organic matter. Eur. J. Soil Sci., 54, 219-

236, https://doi.org/10.1046/j.1365-2389.2003.00544.x, 2003.

Kaiser, K., and Guggenberger, G.: The role of DOM sorption to mineral surfaces in the preservation of

organic matter in soils. Org. Geochem., 31, 711-725, https://doi.org/10.1016/S0146-

6380(00)00046-2, 2000.

Kaiser, K., and Zech, W.: Sorption of dissolved organic nitrogen by acid subsoil horizons and individual

mineral phases. Eur. J. Soil Sci., 51, 403-411, https://doi.org/10.1046/j.1365-2389.2000.00320.x,

2000.

Kaiser, M., Ellerbrock, R.H., Wulf, M., Dultz, S., Hierath, C., and Sommer, M.: The influence of mineral

characteristics on organic matter content, composition, and stability of topsoils under long-term

arable    and    forest    land    use.    J.    Geophys.    Res.-Biogeo.,    117(G2),

https://doi.org/10.1029/2011JG001712, 2012.

Kaiser, M., Wirth, S., Ellerbrock, R.H., and Sommer, M.: Microbial respiration activities related to

sequentially separated, particulate and water-soluble organic matter fractions from arable and forest

topsoils. Soil Biol. Biochem., 42(3): 418-428, https://doi.org/10.1016/j.soilbio.2009.11.018, 2010.



Kleber, M., Sollins, P., and Sutton, R.: A conceptual model of organo-mineral interactions in soils: self-

assembly of organic molecular fragments into zonal structures on mineral surfaces.

Biogeochemistry, 85, 9-24, https://doi.org/10.1007/s10533-007-9103-5, 2007.

Kögel-Knabner, I., and Kleber, M.: Mineralogical, physicochemical, and microbiological controls on

soil organic matter stabilization and turnover, in: Handbook of Soil Sciences Resource Management

and Environmental Impacts, 2nd edn., edited by: Huang, P.M., Li, Y., and Sumner, M.E., CRC Press

Taylor and Francis Group, Boca Raton/London/New York, 7-1 – 7-22, 2011.

Kögel-Knabner, I., Guggenberger, G., Kleber, M., Kandeler, E., Kalbitz, K., Scheu, S., Eusterhues, K.,

and Leinweber, P.: Organo-mineral associations in temperate soils: Integrating biology, mineralogy,

and    organic    matter    chemistry.    J.    Plant.    Nutr.    Soil    Sci.,    171,    61-82,

https://doi.org/10.1002/jpln.200700048, 2008.

Krull, E.S., Baldock, J.A., and Skjemstad, J.O.: Importance of mechanisms and processes of the

stabilization of soil organic matter for modelling carbon turnover. Funct. Plant Biol., 30, 207-222,

https://doi.org/10.1071/FP02085, 2003.

Lehmann, J., and Kleber, M.: The contentious nature of soil organic matter. Nature, 528, 60-68,

http://www.nature.com/doifinder/10.1038/nature16069, 2015.

Lehmann, J., Kinyangi, J., and Solomon, D.: Organic matter stabilization in soil microaggregates:

implications from spatial heterogeneity of organic carbon contents and carbon forms.

Biogeochemistry, 85(1): 45-57, https://doi.org/10.1007/s10533-007-9105-3, 2007.




Leifeld, J., and Kögel-Knabner, I.: Soil organic matter fractions as early indicators for carbon stock
changes under different land-use? Geoderma, 124, 143-155,
https://doi.org/10.1016/j.geoderma.2004.04.009, 2005.
Lorenz, K., and Lal, R.: The depth distribution of soil organic carbon in relation to land use and
management and the potential of carbon sequestration in subsoil horizons. Adv. Agron., 88, 35-66,
https://doi.org/10.1016/S0065-2113(05)88002-2, 2005.
Luo, Z., Wang, E., and Sun, O.J.: Soil carbon change and its responses to agricultural practices in
Australian agro-ecosystems: a review and synthesis. Geoderma, 155, 211-223,
https://doi.org/10.1016/j.geoderma.2009.12.012, 2010.
Margenot, A.J., Calderón, F.J., Bowles, T.M., Parikh S.J., and Jackson, L.E.: Soil organic matter
functional group composition in relation to organic carbon, nitrogen, and phosphorus fractions in
organically managed tomato fields. Soil Sci. Soc. Am. J., 79, 772-782,
https://doi.org/10.2136/sssaj2015.02.0070, 2015.
McDonagh, J.F., Thomsen, T.B., and Magid, J.: Soil organic matter decline and compositional change
associated with cereal cropping in Southern Tanzania. Land Degrad. Dev., 12, 13-26,
https://doi.org/10.1002/ldr.419, 2001.
McKeague, J.: An evaluation of 0.1 M pyrophosphate and pyrophosphate-dithionite in comparison with
oxalate as extractants of the accumulation products in Podzols and some other soils. Can. J. Soil
Sci., 47, 95-99, https://doi.org/10.4141/cjss67-017, 1967.





Mehra, O., and Jackson, M.: Iron oxide removal from soils and clays by a dithionite-citrate system
buffered with sodium bicarbonate. Clays and clay minerals: proceedings of the Seventh National
Conference, 317-327, https://doi.org/10.1016/B978-0-08-009235-5.50026-7, 1958.
Mikutta, R., Schaumann, G.E., Gildemeister, D., Bonneville, S., Kramer, M.G., Chorover, J., Chadwick,
O., and Guggenberger, G.: Biogeochemistry of mineral-organic associations across a long-term
mineralogical soil gradient (0.3-4100kyr), Hawaiian Islands. Geochim. Cosmochim. Acta, 73, 2034-
2060, https://doi.org/10.1016/j.gca.2008.12.028, 2009.
Ontl, T.A., Cambardella, C.A., Schulte, L.A., and Kolka, R.K.: Factors influencing soil aggregation and
particulate organic matter responses to bioenergy crops across a topographic gradient. Geoderma,
255, 1-11, https://doi.org/10.1016/j.geoderma.2015.04.016, 2015.
Parfitt, R.L., Theng, B.K.G., Whitton, J.S., and Shepherd, T.G.: Effects of clay minerals and land use
on organic matter pools. Geoderma, 75, 1-12, https://doi.org/10.1016/S0016-7061(96)00079-1,

1997.

Paul, E.A., Collins, H.P., and Leavitt, S.W.: Dynamics of resistant soil carbon of Midwestern
agricultural soils measured by naturally occurring [14]C abundance. Geoderma, 104, 239-256,
https://doi.org/10.1016/S0016-7061(01)00083-0, 2001.
Poeplau, C., Don, A., Vesterdal, L., Leifeld, J., Van Wesemael, B., Schumacher, J., and Gensior, A.:
Temporal dynamics of soil organic carbon after land-use change in the temperate zone - carbon
response functions as a model approach. Global Change Biol., 17, 2415-2427,
https://doi.org/10.1111/j.1365-2486.2011.02408.x, 2011.



Rabbi, S.M.F., Tighe, M., Cowie, A., Wilson, B.R., Schwenke, G., Mcleod, M., Badgery, W., and
Baldock, J.:. The relationships between land uses, soil management practices, and soil carbon
fractions in South Eastern Australia. Agric. Ecosyst. Environ., 197, 41-52,
https://doi.org/10.1016/j.agee.2014.06.020, 2014.
Rumpel, C., and Kögel-Knabner, I.: Deep soil organic matter-a key but poorly understood component
of terrestrial C cycle. Plant Soil, 338, 143-158, https://doi.org/10.1007/s11104-010-0391-5, 2011.
Rumpel, C., Chaplot, V., Chabbi, A., Largeau, C., and Valentin, C.: Stabilisation of HF soluble and HCl
resistant organic matter in tropical sloping soils under slash and burn agriculture. Geoderma, 145,
347-354, https://doi.org/10.1016/j.geoderma.2008.04.001, 2008.
Schrumpf, M., Kaiser, K., Guggenberger, G., Persson, T., Kögel-Knabner, I., and Schulze, E.D.: Storage
and stability of organic carbon in soils as related to depth, occlusion within aggregates, and
attachment to minerals. Biogeosciences, 10, 1675-1691, https://doi.org/10.5194/bg-10-1675-2013,

2013.

Schwertmann, U., and Cornell, R.M.: Iron oxides in the laboratory, preparation and characterization.
VCH Publication, Weinheim, Germany, 1991.
Schwertmann, V.U.: The differentiation of iron oxide in soils by a photochemical extraction with acid
ammonium oxalate. Zeitschrift für Pflanzenernährung und Bodenkunde, 105, 194-201,
https://doi.org/788 10.1002/jpln.3591050303, 1964.



Shang, C., and Tiessen, H.: Carbon turnover and carbon-13 natural abundance in organo-mineral

fractions of a tropical dry forest soil under cultivation. Soil Sci. Soc. Am. J., 64, 2149-2155,

https://doi.org/10.2136/sssaj2000.6462149x, 2000.

Six, J., Conant, R.T., Paul, E.A., and Paustian, K.: Stabilization mechanisms of soil organic matter:

implications     for     C-saturation     of     soils.     Plant     Soil,     241,     155-176,

https://doi.org/10.1023/A:1016125726789, 2002.

Six, J., Elliott, E.T., Paustian, K., and Doran, J.W.: Aggregation and soil organic matter accumulation

in   cultivated   and   native   grassland   soils.   Soil   Sci.   Soc.   Am.   J.,   62,   1367-1377,

https://doi.org/10.2136/sssaj1998.03615995006200050032x, 1998.

Sleutel, S., Leinweber, P., Van Ranst, E., Kader, M.A., and Jegajeevagan, K.: Organic matter in clay

density fractions from sandy cropland soils with differing land-use history. Soil Sci. Am. J,

75(2), 521-532, https://doi:10.2136/sssaj2010.0094, 2011.

Sollins, P., Glassman, C., Paul, E.A., Swanston, C., Lajtha, K., Heil, J.W., and Elliott, E.A.: Soil carbon

and nitrogen: pools and fractions, in: Standard Soil Methods for Long-Term Ecological Research,

edited by: Robertson, G.P., Bledsoc, C.S., Coleman, D.C., and Sollins, P., Oxford Univ. Press, New

York, 89-105, 1999.

Sollins, P., Kramer, M.G., Swanston, C., Lajtha, K., Filley, T., Aufdenkampe, A.K., Wagai, R., and

Bowden, R.D.: Sequential density fractionation across soils of contrasting mineralogy: evidence for

both microbial- and mineral-controlled soil organic matter stabilization. Biogeochemistry, 96, 209-

231, https://doi.org/10.1007/s10533-009-9359-z, 2009.



Sollins, P., Swanston, C., Kleber, M., Filley, T., Kramer, M., Crow, S., Caldwell, B., Lajtha, K., and

Bowden, R.: Organic C and N stabilization in a forest soil: evidence from sequential density

fractionation. Soil Biol. Biochem., 38, 3313-3324, https://doi.org/10.1016/j.soilbio.2006.04.014,

2006.

Spielvogel, S., Prietzel, J., and Kögel-Knabner, I.: Soil organic matter stabilisation in acidic forest soils

is preferential and soil type-specific. Eur. J. Soil Sci., 59, 74-692, https://doi.org/10.1111/j.1365-

2389.2008.01030.x, 2008.

Stuiver, M., and Polach, H.A.: Discussion: reporting of $^{14}$C data. Radiocarbon, 19, 355-63,

https://doi.org/10.1017/S0033822200003672, 1977.

Tan, Z., Lal, R., Owens, L., and Izaurralde, R.C.: Distribution of light and heavy fractions of soil organic

carbon as related to land use and tillage practice. Soil Till. Res., 92, 53-59,

https://doi.org/10.1016/j.still.2006.01.003, 2007.

Trumbore, S.: Radiocarbon and soil carbon dynamics. Annu. Rev. Earth Pl. Sc., 37, 47-66,

https://doi.org/10.1146/annurev.earth.36.031207.124300, 2009.

Ugawa, S., Miura, S., Iwamoto, K., Kaneko, S., and Fukuda, K.: Vertical patterns of fine root biomass,

morphology and nitrogen concentration in a subalpine fir-wave forest. Plant Soil, 335, 469-478,

https://doi.org/10.1007/s11104-010-0434-y, 2010.

von Lu¨tzow, M., Ko¨gel-Knabner, I., Ekschmitt, K., Flessa, H., Guggenberger, G., Matzner, E., and

Marschner, B.: SOM fractionation methods: relevance to functional pools and to stabilization



mechanisms. Soil Biol. Biochem., 39, 2183-2207, https://doi.org/10.1016/j.soilbio.2007.03.007,

2007.

von Lützow, M., Kögel-Knabner, I., Ekschmitt, K., Matzner, E., Guggenberger, G., Marschner, B., and
Flessa, H.: Stabilization of organic matter in temperate soils: mechanisms and their relevance under
different soil conditions. Eur. J. Soil Sci., 57, 426-44, https://doi.org/10.1111/j.1365-
2389.2006.00809.x, 2006.
Wander, M.M., and Traina, S.J.: Organic matter fractions from organically and conventionally managed
soils: II. Characterization of composition. Soil Sci. Soc. Am. J., 60, 1087-1094,
https://doi.10.2136/sssaj1996.03615995006000040018x, 1996.
Werth, M., and Kuzyakov, Y.: $^{13}$C fractionation at the root-microorganisms-soil interface: a review and
outlook for partitioning studies. Soil Biol. Biochem., 42, 1372-1384,
https://doi.org/10.1016/j.soilbio.2010.04.009, 2010.
Wright, A.L., Dou, F.G., and Hons, F.M.: Crop species and tillage effects on carbon sequestration in
subsurface soil. Soil Sci., 172, 124-131, https://doi.org/10.1097/SS.0b013e31802d11eb, 2007.
Yeasmin, S., Singh, B., Johnston, C.T., and Sparks, D.L.: Evaluation of pre-treatment procedures for
improved interpretation of mid infrared spectra of soil organic matter. Geoderma, 304, 83-92,
https://doi.org/10.1016/j.geoderma.2016.04.008, 2017a.
Yeasmin, S., Singh, B., Johnston, C.T., and Sparks, D.L.: Organic carbon characteristics in density
fractions of soils with contrasting mineralogies. Geochim. Cosmochim. Acta, 218, 215-236,
https://doi.org/10.1016/j.gca.2017.09.007, 2017b.



Yeasmin, S., Singh, B., Kookana, R.S., Farrell, M., Sparks, D., and Johnston, C.T.: Influence of Mineral

characteristics on the retention of low molecular weight organic compounds: a batch sorption-

desorption    and    ATR-FTIR    Study.    J.    Colloid    Interface.    Sci.,    432,    246-257,

https://doi.org/10.1016/j.jcis.2014.06.036, 2014.

Zimmermann, M., Leifeld, J., Abiven, S., Schmidt, M.W., and Fuhrer, J.: Sodium hypochlorite separates

an older soil organic matter fraction than acid hydrolysis. Geoderma, 139, 171-179,

https://doi.org/10.1016/j.geoderma.2007.01.014, 2007.



849        **Figure Captions**

**Fig. 1** Proportion of total OC (a) and total N (b) in the density fractions of four surface (1= 0–
10 cm) and sub-surface (2= 60–70 cm) soils under native and cropped land uses. Density
fractions (DF): POM = <1.8 g cm$^{-1}$ and MOM: 1.8DF = 1.8–2.2 g cm$^{-1}$, 2.2DF = 2.2–2.6 g cm$^{-1}$
$^{1}$ and >2.6DF = >2.6 g cm$^{-1}$. The numbers on the top of the columns represent total recovery
after sequential density fractionation
**Fig. 2** Changes in OC (a) and N (b) concentrations and C:N ratio (c) in density fractions of four
surface (1= 0–10 cm) and sub-surface (2= 60–70 cm) soils with land use conversion from native
to cropped. Density fractions (DF): POM = <1.8 g cm$^{-1}$ and MOM: 1.8DF = 1.8–2.2 g cm$^{-1}$,
2.2DF = 2.2–2.6 g cm$^{-1}$ and >2.6DF = >2.6 g cm$^{-1}$. Error bars represent S.E. of two replicates





**Fig. 3** Change in isotopic values, i.e., $\Delta\delta^{13}$C (a) and $\Delta\delta^{15}$N (b) in density fraction of four surface
(1= 0–10 cm) and sub-surface (2= 60–70 cm) soils with land use change. Density fractions
(DF): POM = <1.8 g cm$^{-1}$ and MOM: 1.8DF = 1.8–2.2 g cm$^{-1}$, 2.2DF = 2.2–2.6 g cm$^{-1}$ and
>2.6DF = >2.6 g cm$^{-1}$. Error bars represent S.E. of the two replicates. Change ($\Delta$) = cropped –
native
**Fig. 4** Relation of $^{14}$C content with $\delta^{13}$C (a) and $\delta^{15}$N (b) in the MOM fraction (2.2DF = 2.2–
2.6 g cm$^{-3}$) of cropped soils (surface + sub-surface)
**Fig. 5** Relation of mineral associated (MOM)-OC loss [% loss = (cropped-native) /native ×100
from 1.8DF, 2.2DF and >2.6DF] with land use conversion with oxalate (ox) and DCB
extractable Fe + Al oxides of cropped soils in surface (a: 0–10 cm) and sub-surface (b: 60–70
cm). r = Pearson's correlation coefficient and * $p$ <0.05 of the correlations





**Table 1** General characteristics of the four bulk soils (<2 mm) from two depths of paired sites (native and cropped) at four locations in NSW, Australia

| WRB soil order | Land uses | Depth (cm) | pH (1:5 H$_2$O) | EC (1:5 H$_2$O) (dS m$^{-1}$) | CEC (mmol$_c$ kg$^{-1}$) | Sand | Silt (%) | Clay |
|---|---|---|---|---|---|---|---|---|
| Ferralsol | Native | 0-10 | 5.6 | 0.19 | 96 | 26 | 43 | 31 |
| | | 60-70 | 5.5 | 0.09 | 69 | 16 | 28 | 56 |
| | Cropped | 0-10 | 5.6 | 0.09 | 89 | 19 | 45 | 36 |
| | | 60-70 | 5.6 | 0.12 | 75 | 13 | 24 | 63 |
| Luvisol | Native | 0-10 | 5.9 | 0.62 | 191 | 40 | 28 | 32 |
| | | 60-70 | 6.6 | 0.32 | 222 | 31 | 16 | 53 |
| | Cropped | 0-10 | 5.0 | 0.75 | 182 | 40 | 31 | 29 |
| | | 60-70 | 6.6 | 0.21 | 216 | 27 | 20 | 53 |
| Vertisol | Native | 0-10 | 6.5 | 0.08 | 213 | 20 | 16 | 64 |
| | | 60-70 | 7.8 | 0.54 | 231 | 16 | 14 | 70 |
| | Cropped | 0-10 | 6.8 | 0.09 | 216 | 23 | 19 | 58 |
| | | 60-70 | 7.7 | 0.27 | 200 | 19 | 15 | 66 |
| Solonetz | Native | 0-10 | 6.0 | 0.13 | 43 | 90 | 4 | 6 |
| | | 60-70 | 6.1 | 0.25 | 50 | 59 | 4 | 37 |
| | Cropped | 0-10 | 5.6 | 0.14 | 36 | 93 | 3 | 4 |
| | | 60-70 | 6.2 | 0.20 | 50 | 56 | 4 | 40 |

All parameters representing mean value of three replicates, except particle size analysis; standard error (S.E.) for pH = 0.00-0.02, EC = <0.01, CEC = 0.1-10.





**Table 2** Semi-quantitative mineralogical composition obtained from XRD analysis of the density fractions of four soils from two depths (surface and sub-surface)

**Surface soils (0-10 cm)\***

| WRB soil order | Density fraction | Phyllosilicates | | | Metal oxide | | | | | Feldspars | | | Quartz |
|---|---|---|---|---|---|---|---|---|---|---|---|---|---|
| | | Kaol | Ill | Sm | Goe | Hem | Gib | Rut | Ant | Mic | Ort | Alb | |
| Ferralsol | POM | x | - | - | tr | x | x | - | - | - | - | - | tr |
| | 1.8DF | xx | x | - | x | xx | xx | - | tr | - | - | - | tr |
| | 2.2DF | xx | - | - | - | xx | xx | - | - | - | - | - | tr |
| | >2.6DF | tr | - | - | x | xxx | x | - | - | - | x | - | x |
| Luvisol | POM | x | x | - | - | - | - | - | - | - | - | x | x |
| | 1.8DF | xx | xx | xxx | - | - | - | - | - | - | - | x | x |
| | 2.2DF | - | xx | x | - | - | - | - | - | - | - | xx | xx |
| | >2.6DF | - | x | - | - | - | - | - | - | - | - | xx | xxx |
| Vertisol | POM | - | x | - | - | - | - | - | - | - | - | - | x |
| | 1.8DF | xx | x | xxx | - | - | - | - | - | - | - | - | xx |
| | 2.2DF | x | - | x | - | - | - | - | - | xx | - | - | xxx |
| | >2.6DF | x | - | - | - | tr | - | - | - | xx | - | xx | xxx |
| Solonetz | POM | x | x | - | - | - | - | - | - | - | - | x | x |
| | 1.8DF | xxx | - | - | - | - | - | tr | tr | - | - | xx | xx |
| | 2.2DF | x | - | - | - | - | - | - | - | - | x | - | xxx |
| | >2.6DF | - | - | - | - | - | - | - | - | - | - | - | xxxx |

**Sub-surface soils (60-70 cm)**

| WRB soil order | Density fraction | Phyllosilicates | | | Metal oxide | | | | Feldspars | | | Quartz |
|---|---|---|---|---|---|---|---|---|---|---|---|---|
| | | Kaol | Ill | Sm | Goe | Hem | Gib | Ant | Mic | Ort | Alb | |
| Ferralsol | POM | x | - | - | x | x | xx | - | - | - | - | tr |
| | 1.8DF | xx | - | - | x | xx | xx | - | - | - | - | tr |
| | 2.2DF | xx | - | - | x | xx | xxx | tr | - | - | - | tr |
| | >2.6DF | x | - | - | - | x | xx | tr | - | x | - | x |
| Luvisol | POM | x | x | - | - | tr | - | - | - | - | x | x |
| | 1.8DF | xxx | xxx | - | - | - | - | - | - | - | - | x |
| | 2.2DF | - | xx | - | - | - | - | - | - | - | xx | xx |
| | >2.6DF | - | x | - | - | - | - | - | - | - | xx | xxx |
| Vertisol | POM | tr | - | x | - | - | - | - | - | - | tr | x |
| | 1.8DF | xx | - | xxx | - | tr | - | - | - | - | - | x |
| | 2.2DF | x | - | x | - | tr | - | x | - | - | x | xx |
| | >2.6DF | - | - | - | - | - | - | - | xx | - | - | xxx |
| Solonetz | POM | - | - | - | x | - | - | x | tr | - | tr | xx |
| | 1.8DF | xx | - | - | x | - | - | x | - | - | - | xx |
| | 2.2DF | xx | x | - | x | - | - | x | - | - | - | xx |
| | >2.6DF | tr | - | - | - | - | - | - | - | - | - | xxxx |

Density fractions: POM = <1.8 g cm⁻³ and MOM: 1.8DF = 1.8-2.2 g cm⁻³, 2.2DF = 2.2-2.6 g cm⁻³ and >2.6DF = >2.6 g cm⁻³. Mineral abbreviations used: Kaol = kaolinite, Ill = illite, Sm =
smectite, Goe = goethite, Hem = hematite, Gib = gibbsite, Rut = rutile, Ant = anatase, Mic = microcline, Ort = orthoclase, Pla = plagioclase, Alb = albite. Estimated proportion of mineral: xxxx
= dominant (>60%), xxx = large (40-60%), xx = moderate (20-40%), x = small (5-20%), tr = trace (<5%), - = non- detectable. *Yeasmin et al. (2017b)





**Table 3** Mean values (n=2) of organic carbon (OC) and nitrogen (N) concentrations, C:N ratios, $\delta^{13}C$, $\delta^{15}N$ and $^{14}C$ activity in the bulk and density fractions of four soils (surface and sub-surface) from native and cropped sites

| WRB soil order | Depth (cm) | Density fraction | OC (g kg⁻¹) | | N (g kg⁻¹) | | C:N | | $\delta^{13}C$ (‰) | | $\delta^{15}N$ (‰) | | $^{14}C$ (pMC) |
|---|---|---|---|---|---|---|---|---|---|---|---|---|---|
| | | | Native | Cropped | Native | Cropped | Native | Cropped | Native | Cropped | Native | Cropped | Cropped |
| Ferralsol | 0–10 | Bulk | 63 | 46 | 6 | 5 | 11 | 10 | -25.9 | -22.3 | 9.7 | 6.2 | 103.9 |
| | | POM | 355 | 285 | 19 | 18 | 19 | 16 | -27.1 | -26.7 | 6.6 | 2.8 | |
| | | 1.8DF | 223 | 165 | 14 | 14 | 16 | 12 | -27.2 | -25.5 | 7.2 | 3.8 | |
| | | 2.2DF | 90 | 45 | 11 | 5 | 8 | 8 | -26.7 | -21.9 | 8.4 | 7.0 | 108.1 |
| | | >2.6DF | 38 | 35 | 4 | 2 | 10 | 18 | -25.5 | -21.2 | 9.8 | 9.4 | |
| | 60–70 | Bulk | 13 | 12 | 1 | 0.9 | 13 | 14 | -24.6 | -22.9 | 8.9 | 9.1 | 60.2 |
| | | POM | 409 | 328 | 5 | 4 | 90 | 87 | -25.6 | -25.9 | 6.4 | 6.3 | |
| | | 1.8DF | 241 | 197 | 5 | 3 | 45 | 63 | -25.8 | -25.4 | 5.6 | 6.2 | |
| | | 2.2DF | 13 | 13 | 1 | 1 | 11 | 13 | -24.4 | -22.7 | 9.0 | 9.3 | 74.0 |
| | | >2.6DF | 9 | 9 | 0.8 | 0.7 | 12 | 12 | -23.9 | -21.0 | 9.7 | 10.8 | |
| Luvisol | 0–10 | Bulk | 26 | 21 | 3 | 2 | 10 | 10 | -22.2 | -23.1 | 8.7 | 7.5 | 104.5 |
| | | POM | 240 | 228 | 14 | 14 | 17 | 17 | -22.7 | -24.2 | 8.1 | 6.2 | |
| | | 1.8DF | 110 | 101 | 9 | 8 | 12 | 12 | -23.1 | -23.1 | 8.5 | 6.5 | |
| | | 2.2DF | 14 | 10 | 2 | 2 | 7 | 6 | -21.4 | -21.9 | 10.8 | 9.9 | 103.8 |
| | | >2.6DF | 3 | 3 | 0.7 | 0.9 | 4 | 3 | -21.7 | -22.4 | 9.9 | 8.9 | |
| | 60–70 | Bulk | 8 | 8 | 1 | 1 | 9 | 7 | -18.2 | -18.4 | 5.8 | 6.0 | 81.3 |
| | | POM | 357 | 331 | 15 | 14 | 24 | 23 | -21.5 | -23.0 | 6.9 | 5.2 | |
| | | 1.8DF | 98 | 61 | 4 | 4 | 24 | 15 | -19.5 | -19.5 | 5.0 | 8.3 | |
| | | 2.2DF | 5 | 3 | 1 | 1 | 5 | 3 | -18.4 | -18.2 | 10.8 | 11.6 | 78.4 |
| | | >2.6DF | 5 | 4 | 0.7 | 0.7 | 6 | 5 | -20.5 | -19.7 | 10.8 | 11.9 | |

Density fractions: POM = <1.8 g cm⁻³; 1.8DF = 1.8-2.2 g cm⁻³ and MOM: 1.8DF = 1.8-2.2 g cm⁻³, 2.2DF = 2.2-2.6 g cm⁻³ and >2.6DF = >2.6 g cm⁻³. pMC = percent modern C. S.E. of the bulk soils: OC (0-0.6), N (0-0.1), C: N (0-0.7), δ¹³C and δ¹⁵N (0-0.02); POM: OC (0.1-3.7), N (0-1.2), C: N (0.1-1.4), δ¹³C (0-0.1) and δ¹⁵N (0.01); 1.8DF: OC (0.1-1.7), N (0-0.4), C: N (0-0.6), δ¹³C (0-0.04) and δ¹⁵N (0-0.02); 2.2DF: OC (0-2.2), N (0-.4), C: N (0-1.1), δ¹³C (0-0.03) and δ¹⁵N (0-4); >2.6DF: OC (0-0.7), N (0-0.1), C: N (0-1.2), δ¹³C (0-0.1) and δ¹⁵N (0-0.01). ±1σ uncertainty of pMC = 0.19-0.28.





**Table 3 (continued)**

| WRB soil order | Depth (cm) | Density fraction | OC (g kg$^{-1}$) Native | OC (g kg$^{-1}$) Cropped | N (g kg$^{-1}$) Native | N (g kg$^{-1}$) Cropped | C:N Native | C:N Cropped | $\delta^{13}$C (‰) Native | $\delta^{13}$C (‰) Cropped | $\delta^{15}$N (‰) Native | $\delta^{15}$N (‰) Cropped | $^{14}$C (pMC) Cropped |
|---|---|---|---|---|---|---|---|---|---|---|---|---|---|
| | 0-10 | Bulk | 17 | 14 | 2 | 1 | 10 | 13 | -22.0 | -21.1 | 4.4 | 6.5 | 99.7 |
| | | POM | 335 | 325 | 11 | 8 | 32 | 43 | -23.2 | -22.3 | 1.7 | 3.5 | |
| | | 1.8DF | 92 | 86 | 6 | 5 | 15 | 16 | -22.5 | -21.3 | 4.9 | 6.1 | |
| | | 2.2DF | 9 | 7 | 2 | 1 | 5 | 8 | -21.9 | -20.4 | 8.6 | 10.4 | 101.4 |
| | | >2.6DF | 6 | 4 | 0.5 | 0.5 | 12 | 8 | -21.7 | -20.6 | 7.8 | 10.1 | |
| Vertisol | 60-70 | Bulk | 10 | 7 | 0.7 | 0.6 | 14 | 12 | -19.7 | -16.7 | 6.7 | 7.3 | 65.0 |
| | | POM | 333 | 262 | 8 | 8 | 44 | 34 | -22.2 | -21.0 | 3.7 | 3.9 | |
| | | 1.8DF | 56 | 47 | 3 | 3 | 19 | 17 | -18.9 | -17.9 | 9.2 | 8.9 | |
| | | 2.2DF | 8 | 8 | 0.9 | 0.8 | 9 | 9 | -17.9 | -16.2 | 14.1 | 14.4 | 62.7 |
| | | >2.6DF | 3 | 2 | 0.2 | 0.3 | 18 | 7 | -15.4 | -11.7 | 12.2 | 13.8 | |
| | 0-10 | Bulk | 15 | 13 | 1 | 1 | 13 | 12 | -24.2 | -22.6 | 3.5 | 4.5 | 106.7 |
| | | POM | 302 | 277 | 16 | 16 | 19 | 17 | -25.7 | -21.1 | 2.9 | 2.6 | |
| | | 1.8DF | 127 | 116 | 11 | 11 | 12 | 11 | -23.8 | -22.5 | 4.8 | 4.2 | |
| | | 2.2DF | 11 | 9 | 2 | 1 | 7 | 8 | -23.2 | -21.6 | 6.5 | 6.3 | 107.4 |
| | | >2.6DF | 1 | 0.4 | 0.1 | 0.1 | 10 | 6 | -24.7 | -23.2 | 1.7 | 9.1 | |
| Solonetz | | Bulk | 3 | 2 | 0.3 | 0.3 | 9 | 7 | -22.7 | -20.5 | 7.4 | 8.8 | 75.6 |
| | | POM | 171 | 125 | 5 | 4 | 36 | 32 | -25.5 | -25.4 | 4.1 | 3.4 | |
| | 60-70 | 1.8DF | 55 | 29 | 4 | 2 | 15 | 13 | -24.6 | -23.9 | 5.4 | 7.7 | |
| | | 2.2DF | 4 | 2 | 0.7 | 0.8 | 6 | 3 | -22.7 | -21.0 | 9.6 | 14.0 | 107.8 |
| | | >2.6DF | 1 | 1 | 0.1 | 0.1 | 10 | 8 | -23.8 | -21.5 | 7.5 | 12.0 | |

Density fractions: POM = <1.8 g cm$^{-3}$ and MOM: 1.8DF = 1.8-2.2 g cm$^{-3}$, 2.2DF = 2.2-2.6 g cm$^{-3}$ and >2.6DF = >2.6 g cm$^{-3}$. pMC = percent modern C. S.E. of the bulk soils: OC (0-0.6), N (0-
0.1), C: N (0-0.7), $\delta^{13}$C and $\delta^{15}$N (0-0.02); POM: OC (0.1-1.2), N (0-1.2), C: N (0.1-3.7), N (0.1-1.4), $\delta^{13}$C (0-0.1) and $\delta^{15}$N (0.01); 1.8DF: OC (0.1-1.7), N (0-0.4), C: N (0-0.6), $\delta^{13}$C (0-0.04) and $\delta^{15}$N (0-
0.02); 2.2DF: OC (0.2-2.2), N (0-.4), C: N (0-1.1), $\delta^{13}$C (0-0.03) and $\delta^{15}$N (0-4); >2.6DF: OC (0-0.7), N (0-0.1), C: N (0-1.2), $\delta^{13}$C (0-0.1) and $\delta^{15}$N (0.19-0.28).




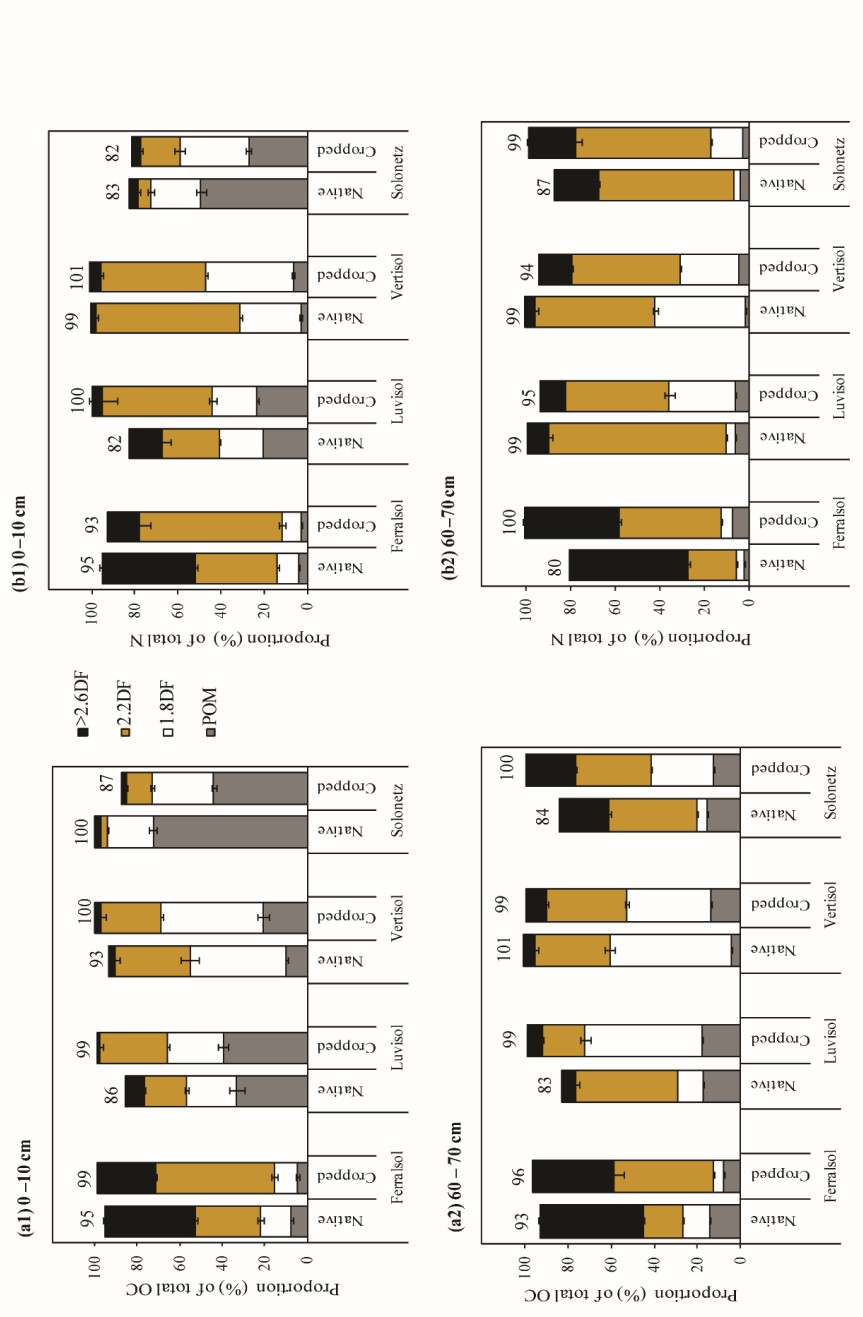

**Figure 1**



**Figure 2**



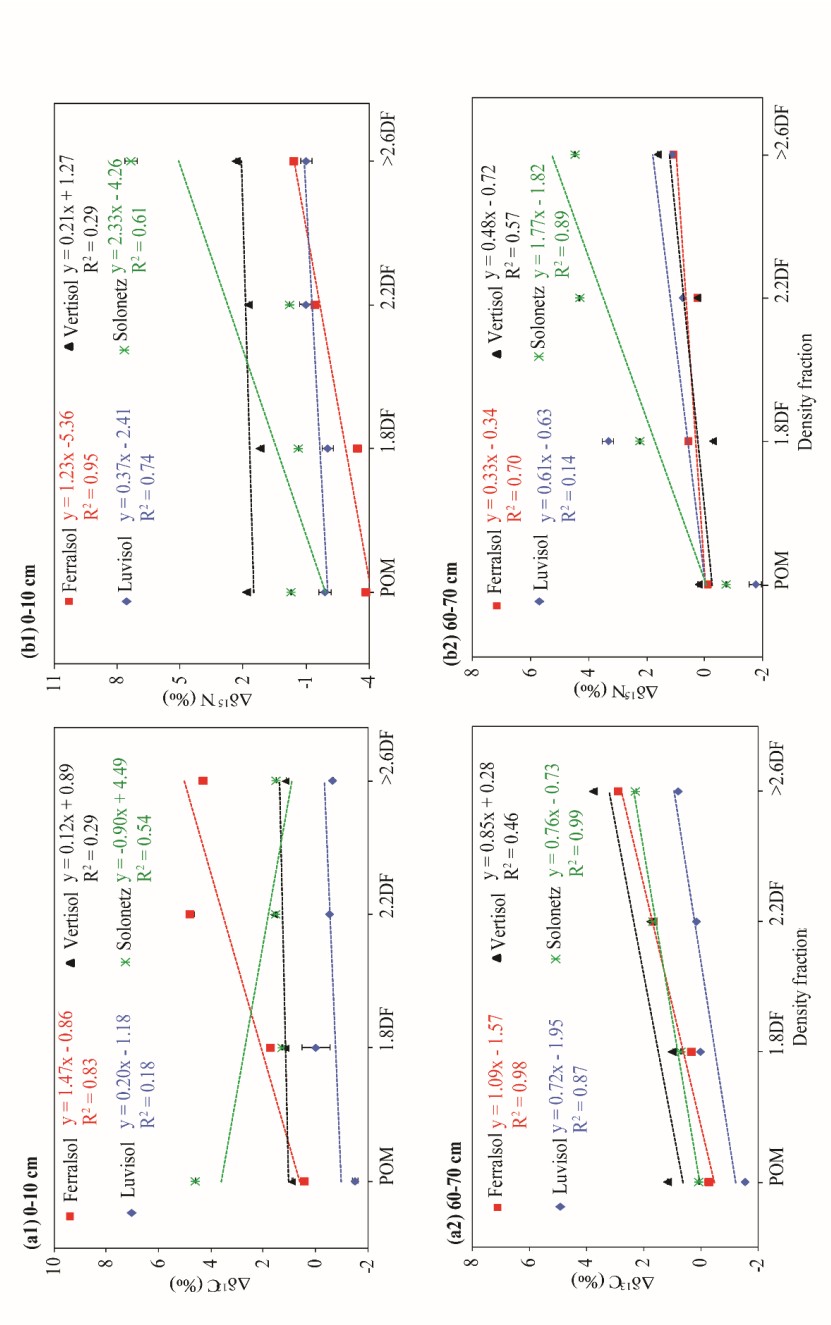

**Figure 3**



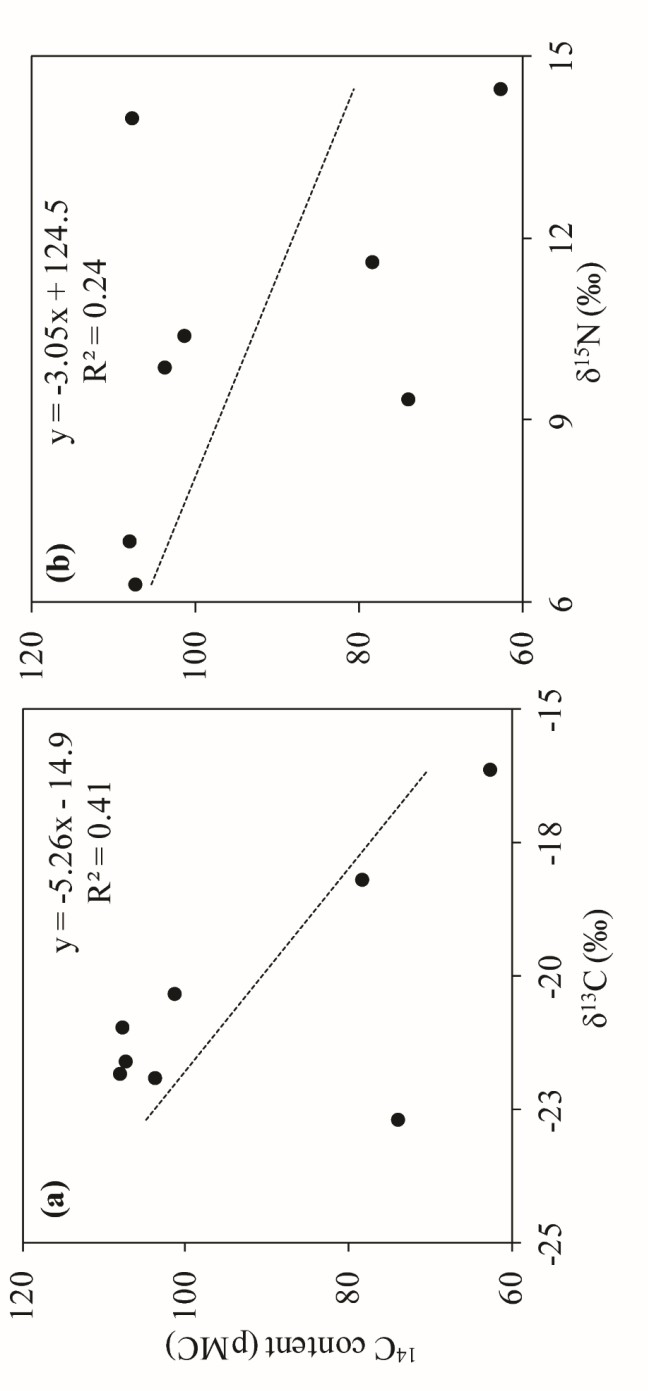

**Figure 4**



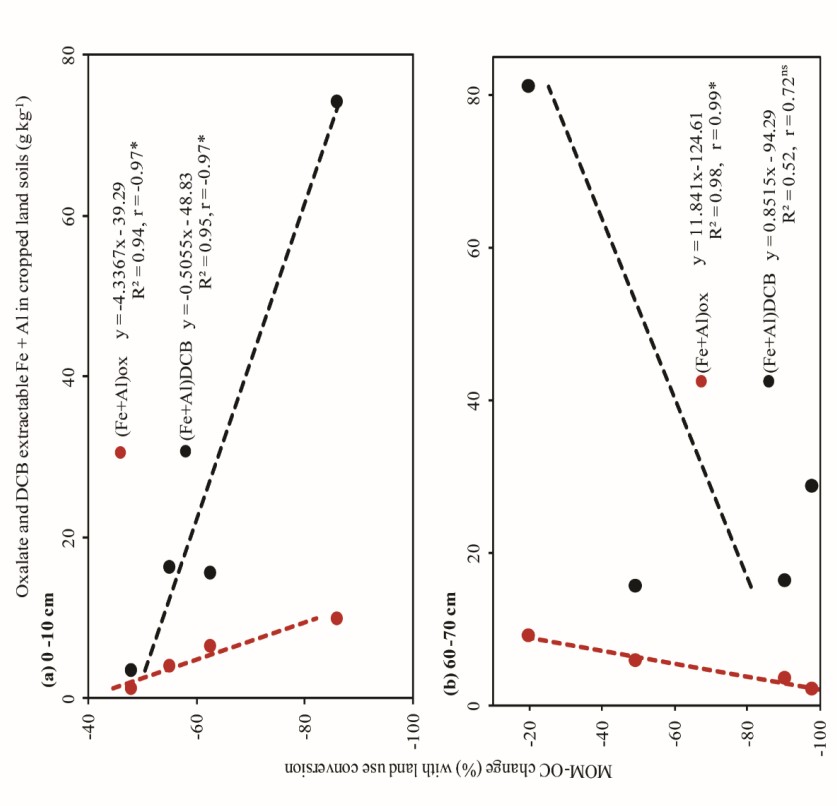

**Figure 5**

