# Peer review of "Changes in Particulate and Mineral Associated Organic Carbon with"

_Biogeosciences, 2019_

## Referee Comment (RC1) · Anonymous Referee #1 · 21 Nov 2019

This manuscript reports changes in the organic matter in different soil types due to land use change, a relevant topic considering the potential of soil C storage in the face of mitigating greenhouse gas emissions. Therefore, undisturbed soil samples were collected at different sites in New South Wales (superficial and subsurface layers), which were determined organic-C and N through the densimetric fractionation (particles size), x-ray diffraction (mineralogy) and isotopic analysis (stable – 13C and 15N; and radioisotope – 14C) sought to point out the associations between organic matter and minerals of different soils. However, there are serious flaws that should be considered. My main concerns are: In a general analysis of the manuscript, the reading is tiring, sometimes excessive in speculations not based on results; and more: what's the question to be answered? - Introduction and objective need to be rewritten

more clearly and cohesively; less descriptive of the methods and paragraphs that best demonstrate the problem studied at work (e.g., LUC impacts on SOM; LUC impacts on different soil types; importance of soil mineralogy on SOM stabilization). - Methods: Site description is poor, but I believe that the most worrying point of this study was the soil sampling strategy. I searched several times for the number of points to form the composite samples, the area coverage or pseudo-repetitions. Thus, results have no statistical validity, especially the absence of error; which culminates in the difficulty of the discussion and conclusions; making the whole work only qualitative and speculative. - I have difficulty understanding the presentation of results. Both soils and land-uses are different between sites; just as the depths have different mineralogical compositions and C-input sources. Sometimes these variables are presented as complementary; others are used comparatively. - Finally, the discussion and conclusion is quite obvious. In this section you could further explore the results, with management suggestions to increase soil carbon stocks and infer about to reaching C-storage limits in each soil type, contributing to greenhouse gas mitigation.

---

## Author Comment (AC1) · 12 Dec 2019

We thank the Referee #1 for taking time to review our manuscript and appreciate the valuable comments and suggestions. We have addressed the comments in the following sections and in a revised manuscript: Referee #1: This manuscript reports changes in the organic matter in different soil types due to land use change, a relevant topic considering the potential of soil C storage in the face of mitigating greenhouse gas emissions. Therefore, undisturbed soil samples were collected at different sites in New South Wales (superficial and subsurface layers), which were determined organic-C and N through the densimetric fractionation (particles size), x-ray diffraction (mineralogy) and isotopic analysis (stable – 13C and 15N; and radioisotope – 14C) sought to point out the associations between organic matter and minerals of different soils.

[Figure]

However, there are serious flaws that should be considered. My main concerns are: -In a general analysis of the manuscript, the reading is tiring, sometimes excessive in speculations not based on results; and more: what's the question to be answered?

Authors: We have made revision in the manuscript to avoid repetition and lengthy monotonous sentences. Objectives have been re-written to clarify the questions possessed and discussion has been improved based on the result sections. They all will be addressed in the revised version.

- Introduction and objective need to be rewritten more clearly and cohesively; less descriptive of the methods and paragraphs that best demonstrate the problem studied at work (e.g., LUC impacts on SOM; LUC impacts on different soil types; importance of soil mineralogy on SOM stabilization).

Authors: Introduction and objective have been revised based on the above comments.

- Methods: Site description is poor, but I believe that the most worrying point of this study was the soil sampling strategy. I searched several times for the number of points to form the composite samples, the area coverage or pseudo-repetitions. Thus, results have no statistical validity, especially the absence of error; which culminates in the difficulty of the discussion and conclusions; making the whole work only qualitative and speculative.

Authors: Site description has been revised. Additionally, detailed geological, climatic and land use information for the sites are now provided in supplementary information (S) Table S1.

**We have incorporated the exact number of sampling point to form the composite samples as 'Random bulk soil samples were collected from eight to eleven spots for the two depths: surface (0–10 cm) and sub-surface (60–70 cm) of each of the paired sites.'**

**We already have information about analytical replicates (pseudo-repetition) in different sections, such as- in section: 2.1 general characterisation of bulk soils: 'All soil analyses were performed in triplicate except the particle size analysis where only one replicate was analysed'. in section: 2.2 Sequential density fractionation: 'The whole fractionation process replicated twice'; in section: 2.4 Soil organic carbon, nitrogen and stable isotopic ratio analyses: 'Duplicate samples were analysed and the precisions for total C, total N, $\delta$13C and $\delta$15N'**

**We have already presented the standard error value for bulk (except texture) + fraction properties in the tables as footnote and in the figures as error bar.**

**About the statistical validity and qualitative findings: We understand reviewer's concern about the lack of replication and statistics. We have acknowledged the issue of field replicates in the manuscript with proper reasoning and references (manuscripts based on similar fractionation scheme that are already published in reputed international journals): "The random samples from the corresponding depth were mixed thoroughly to make the composite sample for each of the individual sites, similar protocol has been used in many published studies (e.g., Kaiser et al., 2010, 2012; Lehmann et al., 2007; Sleutel et al., 2011; Sollins et al., 2006, 2009). Admittedly, that a sampling strategy with separate two or three field replications instead of compositing replications at each site would have been advantageous to find out the spatial variability, but we still believe this sampling protocol would not limit the capacity of this study to assess land use effects in contrasting soils (Kaiser et al., 2012; Sollins et al., 2006)." The above cites articles also worked on size/density fractions of soil organic matter and have not used field replicates, and this issue did not limit them to draw a major conclusion. It is important to point out that these are leading articles in this arena. Additionally, the sampled sites are typical and representative soil types of the desired mineral composition; and mineral composition is not expected to vary within field replicates. The composite samples made up of several random samples are expected to truly represent the organic carbon concentration in the soil. The fractionation scheme used in our experiments (and other studies) is very laborious and time consuming. Thus, use**

of a single composite sample that is representative of the soil type and land use is the pragmatic approach. The laboratory replicates were used to take care on variability in the analytical techniques used to characterize the soil fractions which is an acceptable methodology.

- I have difficulty understanding the presentation of results. Both soils and land-uses are different between sites; just as the depths have different mineralogical compositions and C-input sources. Sometimes these variables are presented as complementary; others are used comparatively.

Authors: We would disagree with the reviewer with this comment. We have clearly mentioned in the materials and method that the four selected locations differ from each other only in soil type. Each location has paired sites- native + cropped lands. The paired sites at each location represented similar landscape, position, climatic conditions and major soil characteristics. We also sampled soils from two depths of each site. Thus, the soils (mineralogy) are different between locations; land uses are different between paired sites of each location. We also clearly mentioned in the result section that mineralogy showed some minor differences between the surface and subsurface soil depths of each location. However, we understand that our writings might have created little confusion. Hence, we have revised the result and discussion section to avoid any confusions or difficulties in understanding of our reader.

- Finally, the discussion and conclusion is quite obvious. In this section you could further explore the results, with management suggestions to increase soil carbon stocks and infer about to reaching C-storage limits in each soil type, contributing to greenhouse gas mitigation.

Authors: We agree with this thoughtful suggestion. We have modified these sections accordingly in the revised version.

References: Kaiser, M., Wirth, S., Ellerbrock, R.H.,and Sommer, M.: Microbial respiration activities related to sequentially separated, particulate and water-soluble organic matter fractions from arable and forest topsoils. Soil Biol. Biochem., 42(3): 418-428, https://doi.org/10.1016/j.soilbio.2009.11.018, 2010. Kaiser, M., Ellerbrock, R.H., Wulf, M., Dultz, S., Hierath, C.,and Sommer, M.: The influence of mineral characteristics on organic matter content, composition, and stability of topsoils under long‐term arable and forest land use. J. Geophys. Res.-Biogeo., 117(G2), https://doi.org/10.1029/2011JG001712, 2012. Lehmann, J., Kinyangi, J.,and Solomon, D.: Organic matter stabilization in soil microaggregates: implications from spatial heterogeneity of organic carbon contents and carbon forms. Biogeochemistry, 85(1): 45-57, https://doi.org/10.1007/s10533-007-9105-3, 2007. Sleutel, S., Leinweber, P., Van Ranst, E., Kader, M.A.,and Jegajeevagan, K.: Organic matter in clay density fractions from sandy cropland soils with differing land-use history. Soil Sci. Soc. Am. J, 75(2), 521-532, https://doi:10.2136/sssaj2010.0094, 2011. Sollins, P., Kramer, M.G., Swanston, C., Lajtha, K., Filley, T., Aufdenkampe, A.K., Wagai, R., and Bowden, R.D.: Sequential density fractionation across soils of contrasting mineralogy: evidence for both microbial- and mineral-controlled soil organic matter stabilization. Biogeochemistry, 96, 209-231,https://doi.org/10.1007/s10533-009-9359-z, 2009. Sollins, P., Swanston, C., Kleber, M., Filley, T., Kramer, M., Crow, S., Caldwell, B., Lajtha, K., and Bowden, R.: Organic C and N stabilization in a forest soil: evidence from sequential density fractionation. Soil Biol. Biochem., 38, 3313-3324,https://doi.org/10.1016/j.soilbio.2006.04.014, 2006.

---

## Referee Comment (RC2) · Anonymous Referee #2 · 24 Dec 2019

The paper addresses the impact of land use change on OM pools in soils with different mineralogy, both at surface and sub-surface depths, which is a relevant topic for publication in BG. Bulk soils were separated into four density fractions, which were then analyzed for their mineralogy, OC and N, isotopic signature and 14C.

My biggest concern about this manuscript is the study design. Only 1 plot per soil type and land use was included in the study, and thus only 1 field replicate was analyzed, given that subsamples collected within each plot were pooled. Therefore, no statistical comparison between adjacent plots is possible and no conclusion regarding the effects of land use change on SOM pools can be drawn. In fact, the different trends in SOM pools after LUC among soil types cannot be ascribed only to the differences in mineralogy, but other important aspects can influence the results, e.g. cropping and tillage

practices, OM inputs, climatic conditions, etc., which the manuscript does not take into consideration or does it only marginally.

Additionally, the manuscript is weak in the following key aspect: - Fundamental details of the sampling strategy (n. of samples per plot, sampling methodology, distance between sampling points) and site design (plot size, and distance between paired plots) are missing;

- Lack of information regarding land use and management at the different sites, which are essential for the result interpretation (e.g. former and current crop species, tillage practices and depth at the cropped sites, OM input at each site e.g. fertilization, crop residue input. . . );

- The main research question are not presented clearly both in the abstract and in the main article;

- Results and Discussion sections are not well structured and often difficult to follow due to a lengthy presentation; Discussion section is too speculative, mostly based on results found by former studies and only partly supported by the obtained results.

SPECIFIC COMMENTS:

ABSTRACT

Is there a "diverse OM input" in soils having different mineralogy (see line 27)? OM input quantity and quantity play a relevant role in determining changes in SOM pools after LUC, and differences found at the different sites cannot be attributed only to mineralogy. Other several variables could have contributed to the observed trends (OM inputs, land use and cropping practices, climatic factors. . .), which the paper is not properly addressing.

It is not explained why a shift in the isotopic signatures could have occurred (e.g. shift from C3 to C4 plants?), at line 30-32.

[Figure]

INTRODUCTION

At Line 49 to 51, it is stated that LUC generally lead to a decline in SOC, but this is not correct, see Guo and Gifford, 2002.

In the paragraph on soil density fractionation, POM is defined as "labile" fraction, without further explanation. This is not always the case, check also Von Lützow et al., 2007. Also at line 75, it should be clarified that organo-mineral associations better protect OM from decomposition but the OM is more processed than the POM fraction (check also Lavallee et al., 2019).

The paragraph at line 86-93 is too descriptive and should be better linked with the research questions. Which kind of insights about of SOM pools do you want to obtain, and why?

Additionally, research questions should be presented more clearly and concisely. It should also be clarified why 2 specific soil depth were chosen for analysis (0-10, and 60-70 cm), in relation to the rooting system and tillage depth, and why a certain Australian region was chosen as a study area (is the specific LUC relevant in that region?).

MATERIALS AND METHODS

In the site description, a precise description of the paired sites is missing (plot size, distance between adjacent sites, precise time of LUC in each site, tillage and cropping practices as tillage depth, fertilization etc; crop and tree species/main understorey vegetation.)

Moreover, fundamental details of the soil sampling strategy (n. of samples per plot, sampling methodology, distance between sampling points) are missing. Also, are organic layers present in the native woodland? Have these layers been sampled? At line 122, what does "absolute mineral soil" mean?

Regarding soil fractionation, how the different densities were selected, e.g. 1.8 g cm-3 to separate POM and MOM?

RESULTS

Presentation of results in paragraph 3.1 "general soil characteristics" and 3.2 "mineralogy of density fractions" is lengthy and not linked to research questions. These paragraphs are focused on site characteristics and differences in mineralogy among sites, but not on the effects of LUC, which seemed to be the main research question in the introductory part.

Regarding the obtained results, nothing can be said statistically about observed trends. The phrasing used in the text (e.g. "remarkable", "notable"..) should not be confused with statistical comparison.

Absolute values of C stored in the different fractions and bulk soils are missing, while only proportions and C concentrations are shown. These data would help explaining the OC losses in the mineral soil.

DISCUSSION

Paragraph 4.1 is mainly about the presentation of C/N ratio trends, so the title "general trends of organic carbon..." is not appropriate. Again, in paragraph 4.2 (404-440) a large share of the discussion is about the role of minerals, while the title reports: "effects of vegetation type".

All the discussion section is rather speculative (see line 460-469, 480-490), and most of the discussion misses a clear support from obtained data.

CONCLUSIONS

Conclusions about effects of LUC and mineralogy on OC are not supported by data: no statistical comparison is possible, and no information regarding cropping practices, and generally about land use management are provided, which can have a great influence on the observed results.

C storage in sub-surface layers is mentioned in the conclusion, but this aspect was not

investigated in the current manuscript.

FIGURES AND TABLES

Mention in the Fig. 1 caption the meaning of error bars.

In Fig. 1: I think that it would be useful to normalize the histograms to 100%, otherwise it is difficult to understand the POM and MOM trends.

Fig. 2: in the different graphs (a1 and a2, b1 and b2, c1 and c2) different scales are used, which is confusing.

Fig. 3: you should add also p values of the observed relationship.

Table 3: add the nomenclature oxide, phyllosilicates and quartz presented in the text at line 267 for a better understanding of the table.

TECHNICAL CORRECTIONS

Line 57: "with differing" is incorrect, change to "differing"

Soil types are sometimes presented in a different order in text and figures (e.g. line 271, 292)

REFERENCES

Yeasmin, S., Singh, B., Johnston, C. T., Sparks, D. L., and Hua, Q.: Changes in Particulate and Mineral Associated Organic Carbon with Land Use in Contrasting Soils, Biogeosciences Discuss., https://doi.org/10.5194/bg-2019-416, in review, 2019

Guo, L. B., & Gifford, R. M. (2002). Soil carbon stocks and land use change: a meta analysis. Global change biology, 8(4), 345-360.

Lavallee, J. M., Soong, J. L., & Cotrufo, M. F. (2019). Conceptualizing soil organic matter into particulate and mineral‐associated forms to address global change in the 21st century. Global change biology.

von Lützow, M., Kögel-Knabner, I., Ekschmitt, K., Flessa, H., Guggenberger, G., Matzner, E., & Marschner, B. (2007). SOM fractionation methods: relevance to functional pools and to stabilization mechanisms. Soil Biology and Biochemistry, 39(9), 2183-2207.

———————————————————

---

## Author Comment (AC2) · 1 Jan 2020

We thank the Referee #2 for taking time to review our manuscript and appreciate the valuable comments and suggestions. We have addressed the comments in the following sections and in the revised manuscript:

Anonymous Referee #2 The paper addresses the impact of land use change on OM pools in soils with different mineralogy, both at surface and sub-surface depths, which is a relevant topic for publication in BG. Bulk soils were separated into four density fractions, which were then analyzed for their mineralogy, OC and N, isotopic signature and 14C. My biggest concern about this manuscript is the study design. Only 1 plot per soil type and land use was included in the study, and thus only 1 field replicate was

analyzed, given that subsamples collected within each plot were pooled. Therefore, no statistical comparison between adjacent plots is possible and no conclusion regarding the effects of land use change on SOM pools can be drawn. In fact, the different trends in SOM pools after LUC among soil types cannot be ascribed only to the differences in mineralogy, but other important aspects can influence the results, e.g. cropping and tillage practices, OM inputs, climatic conditions, etc., which the manuscript does not take into consideration or does it only marginally.

Authors: We understand reviewer's concern about the study design, lack of replication and statistical comparison. We have acknowledged the issue of field replicates in the manuscript with proper reasoning and references (manuscripts based on similar fractionation scheme that are already published in reputed international journals): "The random samples from the corresponding depth were mixed thoroughly to make the composite sample for each of the individual sites, similar protocol has been used in many published studies (e.g., Kaiser et al., 2010, 2012; Lehmann et al., 2007; Sleutel et al., 2011; Sollins et al., 2006, 2009). Admittedly, that a sampling strategy with separate two or three field replications instead of compositing replications at each site would have been advantageous to find out the spatial variability, but we still believe this sampling protocol would not limit the capacity of this study to assess land use effects in contrasting soils (Kaiser et al., 2012; Sollins et al., 2006)." The above cites articles also worked on size/density fractions of soil organic matter and have not used field replicates, and this issue did not limit them to draw a major conclusion. It is important to point out that these are leading articles in this arena.

Additionally, the sampled sites are typical and representative soil types of the desired mineral composition; and mineral composition is not expected to vary within field replicates. The composite samples made up of several random samples are expected to truly represent the organic carbon concentration in the soil. The fractionation scheme used in our experiments (and other studies) is very laborious and time consuming. Thus, use of a single composite sample that is representative of the soil type and land

use is the pragmatic approach. The laboratory replicates were used to take care on variability in the analytical techniques used to characterize the soil fractions which is an acceptable methodology. About the fact that the different trends in SOM pools after LUC among soil types cannot be ascribed only to the differences in mineralogy, but other important aspects can influence the results: We have addressed the other aspects based on the above comments in the revised version of the manuscript.

Additionally, the manuscript is weak in the following key aspect: -Fundamental details of the sampling strategy (n. of samples per plot, sampling methodology, distance between sampling points) and site design (plot size, and distance between paired plots) are missing;

Authors: We have incorporated these aspects in the revised manuscript.

- Lack of information regarding land use and management at the different sites, which are essential for the result interpretation (e.g. former and current crop species, tillage practices and depth at the cropped sites, OM input at each site e.g. fertilization, crop residue input);

Authors: We have incorporated this information in the revised manuscript.

- The main research questions are not presented clearly both in the abstract and in the main article;

Authors: Objectives have been re-written to clarify the research questions.

- Results and Discussion sections are not well structured and often difficult to follow due to a lengthy presentation; Discussion section is too speculative, mostly based on results found by former studies and only partly supported by the obtained results.

Authors: We have made revision in the whole manuscript to avoid repetition and lengthy monotonous sentences. Discussion has been improved based on the result sections. They will all be addressed in the revised version.

[Figure]

SPECIFIC COMMENTS:

ABSTRACT -Is there a "diverse OM input" in soils having different mineralogy (see line 27)?

Authors: 'In surface soils, oxides-dominated MOM lost more OC than phyllosilicates-and quartz -dominated MOM, which is attributed to diverse OM input and the extent of OC saturation limit of soils.' Here by "diverse OM input" we meant 'the difference in OM amount and also isotopic signature variation (due to difference in vegetation C3 or C4 and their OM input).

-OM input quantity and quantity play a relevant role in determining changes in SOM pools after LUC, and differences found at the different sites cannot be attributed only to mineralogy. Other several variables could have contributed to the observed trends (OM inputs, land use and cropping practices, climatic factors), which the paper is not properly addressing.

Authors: We have properly addressed these variables in the revised manuscript.

-It is not explained why a shift in the isotopic signatures could have occurred (e.g. shift from C3 to C4 plants?), at line 30-32.

Authors: The reason for this shift is incorporated.

INTRODUCTION

-At Line 49 to 51, it is stated that LUC generally lead to a decline in SOC, but this is not correct, see Guo and Gifford, 2002.

Authors: We have checked the article and modified this paragraph accordingly.

-In the paragraph on soil density fractionation, POM is defined as "labile" fraction, with-out further explanation. This is not always the case, check also Von Lützow et al., 2007.

Authors: We have checked the article and revised this paragraph accordingly.

-Also, at line 75, it should be clarified that organo-mineral associations better protect OM from decomposition but the OM is more processed than the POM fraction (check also Lavallee et al., 2019).

Authors: We have revised this sentence and incorporated this information with reference.

-The paragraph at line 86-93 is too descriptive and should be better linked with the research questions. Which kind of insights about of SOM pools do you want to obtain, and why?

Authors: We have revised this paragraph accordingly.

-Additionally, research questions should be presented more clearly and concisely. It should also be clarified why 2 specific soil depth were chosen for analysis (0-10, and 60-70 cm), in relation to the rooting system and tillage depth, and why a certain Australian region was chosen as a study area (is the specific LUC relevant in that region?).

Authors: Objectives have been re-written to clarify the research questions. All the other aspects- soil depth, sites etc. have also been addressed in the revised manuscript.

MATERIALS AND METHODS

-In the site description, a precise description of the paired sites is missing (plot size, distance between adjacent sites, precise time of LUC in each site, tillage and cropping practices as tillage depth, fertilization etc.; crop and tree species/main understory vegetation.).

Authors: We have incorporated this information in the revised manuscript as much as possible.

-Moreover, fundamental details of the soil sampling strategy (n. of samples per plot,

sampling methodology, distance between sampling points) are missing. Also, are organic layers present in the native woodland? Have these layers been sampled? At line 122, what does "absolute mineral soil" mean?

Authors: We have incorporated this information in the revised manuscript as much as possible. By 'absolute mineral soil' we meant the mineral soil horizon (60-70 cm) where fresh organic matter (from surface) interference is very less compared to the upper horizons, e.g., 20- 30 cm.

-Regarding soil fractionation, how the different densities were selected, e.g. 1.8 g cm-3 to separate POM and MOM?

Authors: These cut points (1.8, 2.2 and 2.6 g cm-3) for separating POM and MOM were selected based on existing literature [Example: Sollins et al. (2006, 2009), Jones and Singh (2014), Yeasmin et al. (2017a, 2017b)].

RESULTS

-Presentation of results in paragraph 3.1 "general soil characteristics" and 3.2 "mineralogy of density fractions" is lengthy and not linked to research questions. These paragraphs are focused on site characteristics and differences in mineralogy among sites, but not on the effects of LUC, which seemed to be the main research question in the introductory part.

Authors: We have revised these two sections and made them more precise. Although these paragraphs are not directly focusing on the effect of LUC, we believe that these are very important relevant information to answer our research question (effect of LUC)- 'to evaluate the effects of LUC on the POM and MOM pools of both surface and subsurface soils with contrasting mineralogies.'

-Regarding the obtained results, nothing can be said statistically about observed trends. The phrasing used in the text (e.g., "remarkable", "notable") should not be confused with statistical comparison.

Authors: We understand reviewer's concern about the lack statistical comparison. To avoid confusion, we have addressed this issue in the manuscript (Materials and Methods) with proper reasoning and references based the leading articles in this arena (who worked on size/density fractions of soil organic matter and have not used field replicates and this issue did not limit them to draw a major conclusion).

-Absolute values of C stored in the different fractions and bulk soils are missing, while only proportions and C concentrations are shown. These data would help explaining the OC losses in the mineral soil.

Authors: The values are added in the supplementary information.

DISCUSSION

-Paragraph 4.1 is mainly about the presentation of C/N ratio trends, so the title "general trends of organic carbon." is not appropriate. Again, in paragraph 4.2 (404-440) a large share of the discussion is about the role of minerals, while the title reports: "effects of vegetation type".

Authors: We have modified the sub-heading titles.

-All the discussion section is rather speculative (see line 460-469, 480-490), and most of the discussion misses a clear support from obtained data.

Authors: Discussion has been improved based on the result sections.

CONCLUSIONS

-Conclusions about effects of LUC and mineralogy on OC are not supported by data: no statistical comparison is possible, and no information regarding cropping practices, and generally about land use management are provided, which can have a great influence on the observed results.

Authors: Conclusions have been modified accordingly.

-C storage in sub-surface layers is mentioned in the conclusion, but this aspect was not investigated in the current manuscript.

Authors: We understand reviewer's concern about this sentence- 'Sub-surface soils can act as a potential C sink for long-term C storage, together with improved land use and vegetation management systems.' We stated this sentence as a possible fact, since, the sub-surface soils lost less OC than surface soils with LUC in our study. However, we have revised this sentence.

FIGURES AND TABLES

-Mention in the Fig. 1 caption the meaning of error bars.

Authors: We have incorporated this information in the caption.

-In Fig. 1: I think that it would be useful to normalize the histograms to 100%, otherwise it is difficult to understand the POM and MOM trends.

Authors: We have modified the figure accordingly.

-Fig. 2: in the different graphs (a1 and a2, b1 and b2, c1 and c2) different scales are used, which is confusing.

Authors: We have modified the scales accordingly.

-Fig. 3: you should add also p values of the observed relationship.

Authors: We will consider this.

-Table 3: add the nomenclature oxide, phyllosilicates and quartz presented in the text at line 267 for a better understanding of the table.

Authors: We will add this information.

TECHNICAL CORRECTIONS

-Line 57: "with differing" is incorrect, change to "differing" Soil types are sometimes

presented in a different order in text and figures (e.g. line 271, 292)

Authors: We have made these corrections accordingly.

REFERENCES Yeasmin, S., Singh, B., Johnston, C. T., Sparks, D. L., and Hua, Q.: Changes in Particulate and Mineral Associated Organic Carbon with Land Use in Contrasting Soils, Biogeosciences Discuss., https://doi.org/10.5194/bg-2019-416, in review, 2019 Guo, L. B., & Gifford, R. M. (2002). Soil carbon stocks and land use change: a meta-analysis. Global change biology, 8(4), 345-360. Lavallee, J. M., Soong, J. L., & Cotrufo, M. F. (2019). Conceptualizing soil organic matter into particulate and mineralâ ËŸA ËĞRassociated forms to address global change in the 21st century. Global change biology. von Lützow, M., Kögel-Knabner, I., Ekschmitt, K., Flessa, H., Guggenberger, G., Matzner, E., & Marschner, B. (2007). SOM fractionation methods: relevance to functional pools and to stabilization mechanisms. Soil Biology and Biochemistry, 39(9), 2183-2207.

Authors: We have incorporated these references in the manuscript.

References: Jones, E., and Singh, B.: Organo-mineral interactions in contrasting soils under natural vegetation. Front. Environ. Sci., 2, 2. https://doi.org/10.3389/fenvs.2014.00002, 2014. Kaiser, M., Ellerbrock, R.H., Wulf, M., Dultz, S., Hierath, C.,and Sommer, M.: The influence of mineral characteristics on organic matter content, composition, and stability of topsoils under long‐term arable and forest land use. J. Geophys. Res.-Biogeo., 117(G2), https://doi.org/10.1029/2011JG001712, 2012. Kaiser, M., Wirth, S., Ellerbrock, R.H.,and Sommer, M.: Microbial respiration activities related to sequentially separated, particulate and water-soluble organic matter fractions from arable and forest topsoils. Soil Biol. Biochem., 42(3): 418-428, https://doi.org/10.1016/j.soilbio.2009.11.018, 2010. Lehmann, J., Kinyangi, J.,and Solomon, D.: Organic matter stabilization in soil microaggregates: implications from spatial heterogeneity of organic carbon contents and carbon forms. Biogeochemistry, 85(1): 45-57, https://doi.org/10.1007/s10533-

007-9105-3, 2007. Sleutel, S., Leinweber, P., Van Ranst, E., Kader, M.A.,and Jegajeevagan, K.: Organic matter in clay density fractions from sandy cropland soils with differing land-use history. Soil Sci. Soc. Am. J, 75(2), 521-532, https://doi:10.2136/sssaj2010.0094, 2011. Sollins, P., Kramer, M.G., Swanston, C., Lajtha, K., Filley, T., Aufdenkampe, A.K., Wagai, R., and Bowden, R.D.: Sequential density fractionation across soils of contrasting mineralogy: evidence for both microbial- and mineral-controlled soil organic matter stabilization. Biogeochemistry, 96, 209-231,https://doi.org/10.1007/s10533-009-9359-z, 2009. Sollins, P., Swanston, C., Kleber, M., Filley, T., Kramer, M., Crow, S., Caldwell, B., Lajtha, K., and Bowden, R.: Organic C and N stabilization in a forest soil: evidence from sequential density fractionation. Soil Biol. Biochem., 38, 3313-3324,https://doi.org/10.1016/j.soilbio.2006.04.014, 2006. Yeasmin, S., Singh, B., Johnston, C.T., and Sparks, D.L.: Evaluation of pre-treatment procedures for improved interpretation of mid infrared spectra of soil organic matter. Geoderma, 304, 83-92, https://doi.org/10.1016/j.geoderma.2016.04.008, 2017a. Yeasmin, S., Singh, B., Johnston, C.T., and Sparks, D.L.: Organic carbon characteristics in density fractions of soils with contrasting mineralogies. Geochim. Cosmochim. Acta, 218, 215-236, https://doi.org/10.1016/j.gca.2017.09.007, 2017b.